# The Nse5/6-like SIMC1-SLF2 complex localizes SMC5/6 to viral replication centers

Martina Oravcová[1], Minghua Nie[1], Nicola Zilio[2], Shintaro Maeda[3], Yasaman Jami-Alahmadi[4], Eros Lazzerini-Denchi[5], James A Wohlschlegel[4], Helle D Ulrich[2], Takanori Otomo[3,6]*, Michael N Boddy[1]*

[1]Department of Molecular Medicine, The Scripps Research Institute, La Jolla, United States; [2]Institute of Molecular Biology, Mainz, Germany; [3]Department of Integrative Structural and Computational Biology, The Scripps Research Institute, La Jolla, United States; [4]Department of Biological Chemistry, David Geffen School of Medicine, University of California, Los Angeles, Los Angeles, United States; [5]Laboratory of Genome Integrity, National Cancer Institute, Bethesda, United States; [6]San Diego Biomedical Research Institute, San Diego, United States

*For correspondence:
totomo@sdbri.org (TO);
nboddy@scripps.edu (MNB)

Competing interest: The authors declare that no competing interests exist.

**Abstract** The human SMC5/6 complex is a conserved guardian of genome stability and an emerging component of antiviral responses. These disparate functions likely require distinct mechanisms of SMC5/6 regulation. In yeast, Smc5/6 is regulated by its Nse5/6 subunits, but such regulatory subunits for human SMC5/6 are poorly defined. Here, we identify a novel SMC5/6 subunit called SIMC1 that contains SUMO interacting motifs (SIMs) and an Nse5-like domain. We isolated SIMC1 from the proteomic environment of SMC5/6 within polyomavirus large T antigen (LT)-induced subnuclear compartments. SIMC1 uses its SIMs and Nse5-like domain to localize SMC5/6 to polyomavirus replication centers (PyVRCs) at SUMO-rich PML nuclear bodies. SIMC1's Nse5-like domain binds to the putative Nse6 orthologue SLF2 to form an anti-parallel helical dimer resembling the yeast Nse5/6 structure. SIMC1-SLF2 structure-based mutagenesis defines a conserved surface region containing the N-terminus of SIMC1's helical domain that regulates SMC5/6 localization to PyVRCs. Furthermore, SLF1, which recruits SMC5/6 to DNA lesions via its BRCT and ARD motifs, binds SLF2 analogously to SIMC1 and forms a separate Nse5/6-like complex. Thus, two Nse5/6-like complexes with distinct recruitment domains control human SMC5/6 localization.

## Editor's evaluation

This paper will be of interest to the chromosome biology field and SMC researchers in particular. The study provides cell biological, biochemical, and structural modeling evidence that a new Nse5-like protein named SIMC1 is a paralog of SLF1, and that the two compete for SLF2-Smc5/6 binding. The authors also show that SIMC1 targets SMC5/6 to polyomavirus replication centers through its SUMO binding motifs (SIMs), while SIMC1 is not recruited to DNA damage sites, supporting a specific role for SIMC1 in Smc5/6 recruitment for viral restriction.

## Introduction

SMC5/6 is the most enigmatic of the three eukaryotic structural maintenance of chromosomes (SMC) complexes, which together maintain genome stability and prevent disease (*Verver et al., 2016*; *Aragón, 2018*; *Uhlmann, 2016*). All SMC complexes contain heterodimeric SMC ATPases that

topologically embrace DNA. With this conserved activity, cohesin (SMC1/3) holds sister chromatids together prior to mitosis, condensin (SMC2/4) compacts chromosomes for mitotic segregation and SMC5/6 resolves DNA recombination intermediates prior to mitosis and meiosis. SMC5/6 recognizes certain DNA structures for example, supercoiled DNA, and can compact such substrates (*Serrano et al., 2020*; *Gutierrez-Escribano et al., 2020*). This function may protect DNA structures from aberrant processing or, present them in a manner that assists their repair (*Agashe et al., 2021*; *Copsey et al., 2013*; *Wehrkamp-Richter et al., 2012*; *Bermúdez-López and Aragon, 2017*; *Bermúdez-López et al., 2016*).

In addition to its canonical role in genome stability, human SMC5/6 is emerging uniquely amongst SMC complexes as a critical viral restriction factor. SMC5/6 inhibits the transcription and/or replication of several viruses including hepatitis B (HBV; *Decorsière et al., 2016*; *Murphy et al., 2016*), herpes simplex (HSV-1; *Xu et al., 2018*), human papillomavirus (HPV; *Bentley et al., 2018*; *Gibson and Androphy, 2020*), Epstein-Barr virus (EBV; *Yiu et al., 2022*), and unintegrated human immunodeficiency virus (HIV-1; *Dupont et al., 2021*). To overcome restriction by SMC5/6, HBV and HIV-1 use the viral proteins HBx or Vpr to target the complex for proteasomal turnover (*Decorsière et al., 2016*; *Dupont et al., 2021*). How SMC5/6 combats viruses remains undefined, but it is known that SMC5/6 collaborates with topoisomerases to silence transcription and replication of both HBV and HSV-1 (*Xu et al., 2018*). Thus, SMC5/6 may bind supercoiled viral genomes and compact them, making them refractory to access by transcription and replication factors.

The 'core' SMC5/6 complex is a well-conserved hexamer, consisting of the SMC5-SMC6 heterodimeric ATPases, the non-SMC-elements 1/3/4 (NSMCE1/3/4 in human and Nse1/3/4 in yeast) heterotrimer that bridges the SMC5/6 head groups, and the SMC5-associated SUMO ligase NSMCE2 (Nse2 in yeast, reviewed in *Aragón, 2018*; *Uhlmann, 2016*; *Oravcová and Boddy, 2019a*; *Palecek, 2018*). In yeast, additional subunits called Nse5 and Nse6 form a heterodimer (Nse5/6) that plays multiple roles in regulating the Smc5/6 core (*Pebernard et al., 2006*; *Duan et al., 2009*; *Palecek et al., 2006*). Nse5/6 recruits and loads Smc5/6 on chromatin (*Oravcová et al., 2019b*; *Etheridge et al., 2021*), and inhibits the ATPase activity of Smc5/6, possibly to stabilize its chromatin association (*Hallett et al., 2021*; *Taschner et al., 2021*). Likely through loading Smc5/6 on DNA and recruiting SUMO, Nse5/6 also 'activates' Nse2, promoting its SUMO ligase activity on chromatin (*Oravcová et al., 2019b*; *Albuquerque et al., 2013*; *Yu et al., 2021*). Smc5/6 recruitment and activation at DNA damage sites is promoted by an interaction between Nse5/6 and the BRCT domain-containing proteins Brc1/Rtt107, which in turn interact with Rad18 and gamma-H2AX at DNA lesions (*Oravcová and Boddy, 2019a*; *Oravcová et al., 2019b*; *Etheridge et al., 2021*; *Leung et al., 2011*). Consistent with defects in Smc5/6 regulation, cells lacking Nse5/6 are hypersensitive to genotoxins and exhibit aberrant chromosome segregation in mitotic and meiotic cells (*Wehrkamp-Richter et al., 2012*; *Oravcová and Boddy, 2019a*; *Pebernard et al., 2006*; *Oravcová et al., 2019b*; *Etheridge et al., 2021*; *Bustard et al., 2012*).

Despite such crucial roles in yeast, Nse5/6 is not evolutionarily conserved at the primary sequence level. In plants, an SMC5/6-associated heterodimer of ASAP1 and SNI1 has been suggested to be the counterpart of Nse5/6 but this was based on its functions and structural modeling (*Yan et al., 2013*). In humans, two proteins SMC5/6 localization factor 1 and 2 (SLF1 and SLF2) were proposed to fulfill the roles of Nse5 and Nse6, respectively (*Räschle et al., 2015*). Consistent with yeast phenotypes, SLF1/2 recruits SMC5/6 to DNA lesions, and its depletion renders cells sensitive to replication stress (*Räschle et al., 2015*). SLF2 is also required for SMC5/6 localization to unintegrated HIV-1 DNA to induce its transcriptional silencing, which surprisingly does not require SLF1 (*Dupont et al., 2021*). SLF2 contains an Nse6-like domain in its C-terminus, detectable by algorithms for picking up remote homologies (e.g. HHPred, *Zimmermann et al., 2018*). But SLF1 bears no sequence similarity to Nse5 (*Räschle et al., 2015*).

The existing SMC5/6 core and regulatory subunits do not explain its recruitment to and antagonism of viruses. Here, we used proximity labeling and protein purification to define the proteomic environment of SMC5/6 within nuclear foci induced by the large T antigen of polyomavirus SV40. In this context, we identified a poorly characterized protein SUMO interacting motifs containing 1 (SIMC1) as a novel SMC5/6 cofactor. SIMC1 concentrates at SUMO-rich PML nuclear bodies (PML NBs) via its SUMO-interacting motifs (SIMs; *Sun and Hunter, 2012*) and interacts with SLF2 via its Nse5-like domain, thereby localizing SMC5/6 to polyomavirus replication centers (PyVRCs). Structural

analyses combining AlphaFold prediction and cryo-electron microscopy (cryo-EM) reveal the Nse5/6-like structure of the SIMC1-SLF2 complex. In addition, structure-based mutational analysis establishes the importance of SIMC1's Nse5-like domain in SMC5/6 regulation. Overall, we find that SIMC1 and SLF1 form distinct Nse5/6-like complexes with SLF2 to direct the antiviral and DNA repair responses of human SMC5/6.

## Results

### SIMC1 identified proximal to SMC5 within LT-induced subnuclear foci

The large T antigen (LT) of SV40 induces nuclear foci in cells that contain DNA replication and repair factors related to the viral lifecycle (*Boichuk et al., 2010*). Moreover, although functionally unexplored, a previous proteomic analysis of LT-associated proteins identified several subunits of SMC5/6 (*Fine et al., 2012*). Consistent with this, we detected SMC5/6 in nuclear foci that colocalized with SV40 LT in HEK293T cells (*Figure 1a*). HEK293 cells lack SMC5/6 foci but expression of LT led to the emergence of puncta containing both SMC5/6 and LT. Thus, SMC5/6 may be a new cellular factor involved in the SV40 lifecycle.

Based on the foregoing, we reasoned that virus-related SMC5/6 cofactors may be concentrated within LT-induced nuclear foci (*Figure 1b*). Therefore, we used proximity labeling with biotin and SILAC-based mass spectrometry to identify proteins in the environment of human SMC5 that was fused to the promiscuous biotin ligase BirA* and expressed in HEK293T cells (*Roux et al., 2012*; *Figure 1c*). We defined the top-ranking hits by the SILAC ratios of proteins recovered in SMC5-BirA* versus the NLS-BirA* control purifications (*Figure 1d*, *Supplementary file 1* and PRIDE database: PXD033923). Proteins on the list provide high confidence in the data, as five of the top nine most enriched SMC5 interactors are known components of the SMC5/6 core complex (*Figure 1d*). In addition, the presumptive Nse6 orthologue SLF2 was detected but the suggested Nse5 counterpart SLF1 was missing. Instead, a poorly characterized protein called SIMC1 (C5orf25) was detected with similar enrichment as SLF2 (*Figure 1d*).

### SIMC1 contains a yeast Nse5-like domain

SIMC1 is a modular protein that contains tandem SIMs within a large N-terminal intrinsically disordered region (IDR) and a predicted alpha-helical rich C-terminal region (*Figure 2a*). Standard sequence searches failed to identify homologous proteins in lower eukaryotes such as yeast. Therefore, we conducted remote homology searches on the HHpred server (*Zimmermann et al., 2018*). Searching against yeast and *A. thaliana* databases returned yeast Nse5 and plant SNI1 as the most probable hits in each species for SIMC1 C-terminal amino acids 467–802 (*Figure 2—figure supplement 1*). Consistent with previous reports (*Räschle et al., 2015*) a search with the putative Nse5 orthologue SLF1 did not identify Nse5. These results led us to hypothesize that SIMC1 is the Nse5-like regulator of human SMC5/6 that directs SMC5/6 to antagonize viral infections.

### SIMC1 and SLF2 interact via their Nse5- and Nse6-like regions

In all species tested, Nse5/6-like complexes are composed of two alpha-helical domain-containing proteins that form an obligate heterodimer. We, therefore, investigated if and how SIMC1 and SLF2 interact. When co-expressed in cells, full-length SIMC1 and SLF2 colocalize in subnuclear foci and specifically co-immunoprecipitate (*Figure 2b*, *Figure 2—figure supplement 2*). Co-expressed SIMC1 and SLF2 are more abundant than when each is expressed with controls, which likely reflects the stabilization of each protein when in complex (*Figure 2—figure supplement 2*).

Next, we sought to determine the interacting regions of SIMC1 and SLF2. For SLF2, we tested a C-terminal construct (SLF2$^{CTD}$; 635–1173) that contains the Nse6-like region previously reported to interact with SLF1 (*Adamus et al., 2020*). This construct supports binding to full-length SIMC1 (*Figure 2c*). For SIMC1, we tested four N-terminal truncations, starting with residues 284, 381, 457 or 652. The results showed that SIMC1 constructs 284 and 381 stably interact with SLF2$^{CTD}$ (*Figure 2c*). However, the shortest construct that lacks the N-terminal part of the SIMC1 Nse5-like region (652) was poorly expressed (*Figure 2c*). Whilst SIMC1 construct 457 interacts with SLF2$^{CTD}$ and contains the entire Nse5-like region defined by HHpred, the binding appears weakened (*Figure 2c*). Thus, the minimum SLF2-interacting region of SIMC1 with full binding capacity is located between residues 381

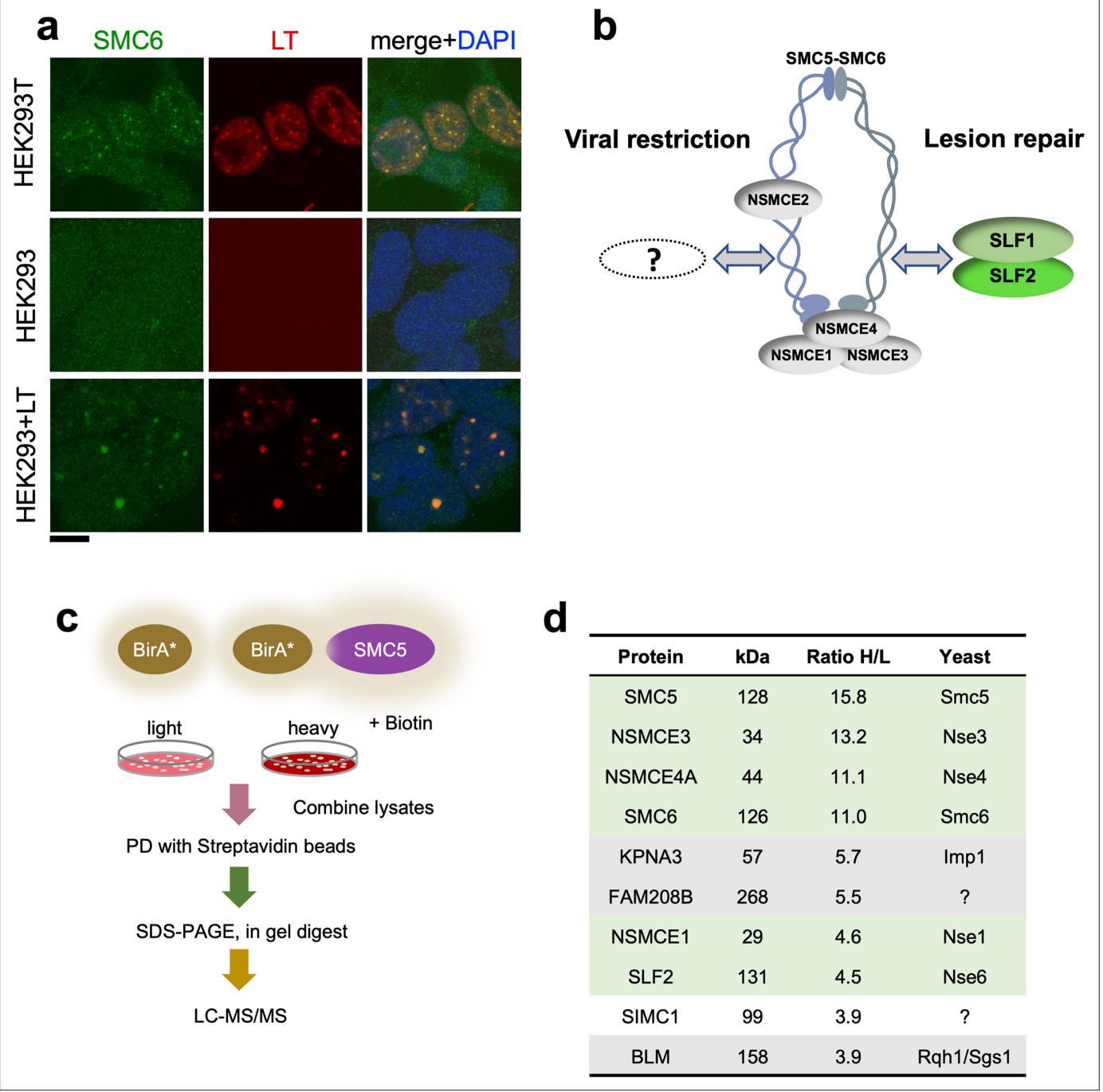

**Figure 1.** SIMC1 detected in proximity of SMC5 in LT-induced foci. (**a**) Representative immunofluorescence images of HEK293T, HEK293 cell lines and HEK293 cells transiently expressing SV40 large T (LT) antigen. Cells were fixed, stained with SMC6 (green), LT (red) antibodies and DAPI (blue). Scale bar 10 µm. (**b**) Schematic of SMC5/6 complex subunits showing the SLF1/SLF2 cofactors recruiting the complex to DNA lesions and depicting the hypothesis of SMC5/6 cofactor involved in SV40 virus lifecycle. (**c**) Experimental design for human SMC5 BioID with SILAC. (**d**) The top SMC5 interacting proteins identified in BioID screen are listed with their ratio of heavy (**H**) to light (**L**) enrichment. Where known, fission and budding yeast orthologues are listed. The green rows contain known SMC5/6 subunits, grey ones are possible novel interactors of human SMC5. BioID datasets are available in the PRIDE database (https://www.ebi.ac.uk/pride/archive) under the accession code PRIDE PXD033923 and in *Supplementary file 1*.

and 872. In parallel, we succeeded in co-purifying an apparently heterodimeric complex of SIMC1[284] and SLF2[CTD] proteins expressed in insect cells (*Figure 2—figure supplement 3*). Thus, the C-terminal domains of SIMC1 and SLF2 directly interact to form a stable complex, supporting our hypothesis that the SIMC1-SLF2 complex is an Nse5/6-like regulatory factor for human SMC5/6.

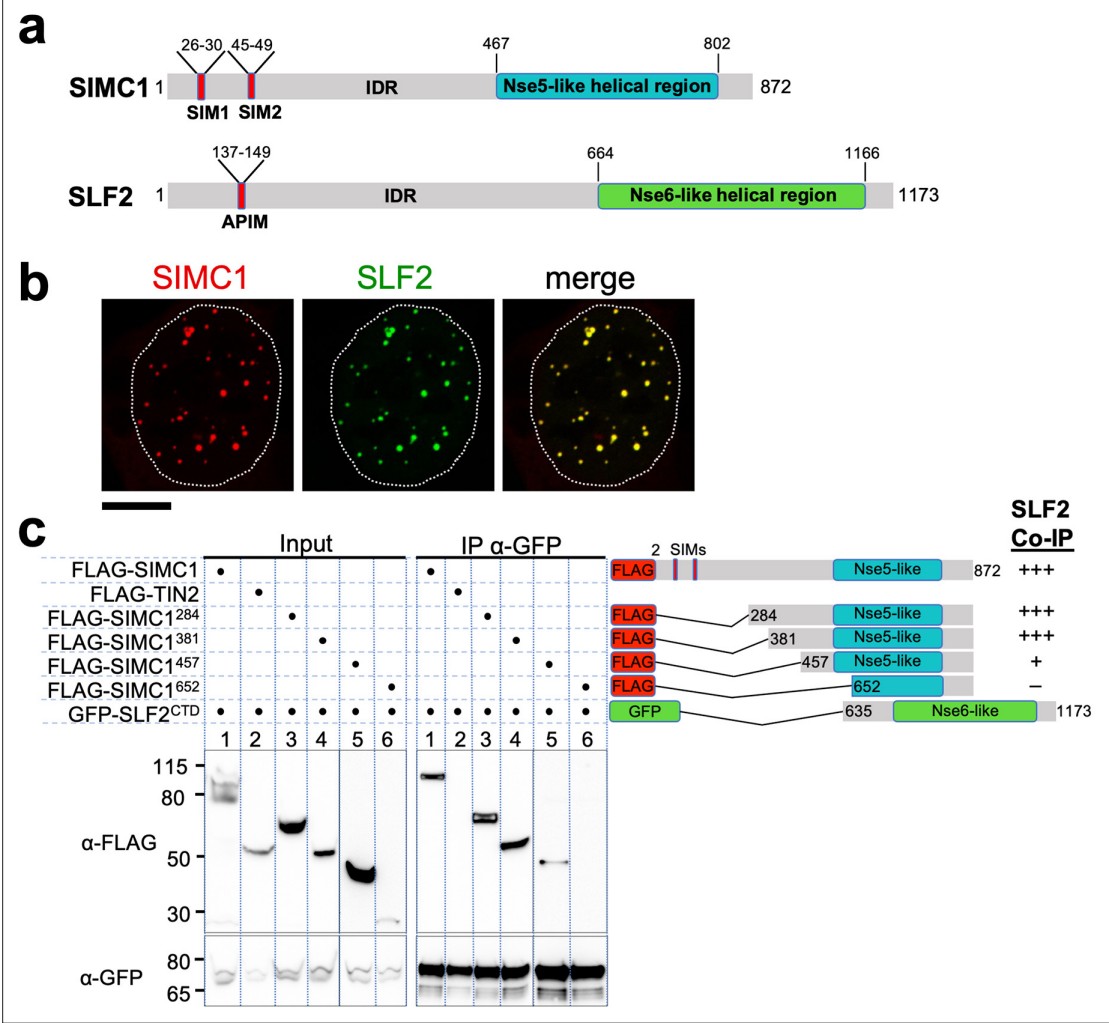

**Figure 2.** The Nse5- and Nse6-like domains of SIMC1 and SLF2 interact. (**a**) Schematic representation of SIMC1 and SLF2. SIM1/2, SUMO interaction motifs; IDR, intrinsically disordered regions; APIM, AlkB homologue 2 PCNA-interacting motif. (**b**) Representative immunofluorescence images of U2OS cells overexpressing mCherry-SIMC1 and GFP-SLF2 that were fixed and stained with DAPI. Dotted outline marks nucleus. Scale bar 10 μm. (**c**) Western blot of GFP-Trap immunoprecipitation from HEK293 cells transfected with plasmids expressing the indicated combinations of proteins. Truncation constructs of SIMC1 and SLF2 are represented schematically including the status of SIMC1-SLF2 interaction that is, +++ > +, − undetectable. Full and unedited blots provided in *Figure 2—source data 1*.

The online version of this article includes the following source data and figure supplement(s) for figure 2:

**Source data 1.** Full and unedited blots corresponding to panel (**c**).

**Figure supplement 1.** HHPred homology searches for full-length SIMC1 (Q8NDZ2) and SLF2 (Q8IX21).

**Figure supplement 2.** Full-length SIMC1 and SLF2 co-immunoprecipitate.

**Figure supplement 2—source data 1.** Full and unedited blots.

**Figure supplement 3.** Purification of the SIMC1$^{284}$-SLF2$^{CTD}$ complex from insect cells.

**Figure supplement 3—source data 1.** Full and unedited gels.

## SIMC1 interacts with SMC5/6 and SUMO pathway factors

To probe the local environment where SIMC1 functions, we used proximity labeling and mass spectrometry in cells stably expressing either Myc-BirA* or Myc-BirA*-SIMC1 together with SLF2. Because the biotinylation radius of BirA* is restricted to ~10 nm, direct protein contacts within multiprotein complexes can be selectively detected in denaturing but not native conditions (*Kim et al., 2014*; *Kim et al., 2016*). Indeed, in native BioID conditions, SLF2 and all known subunits of the SMC5/6 complex were detected (*Figure 3a*) whereas under the denaturing conditions of BioID, SLF2 but no other

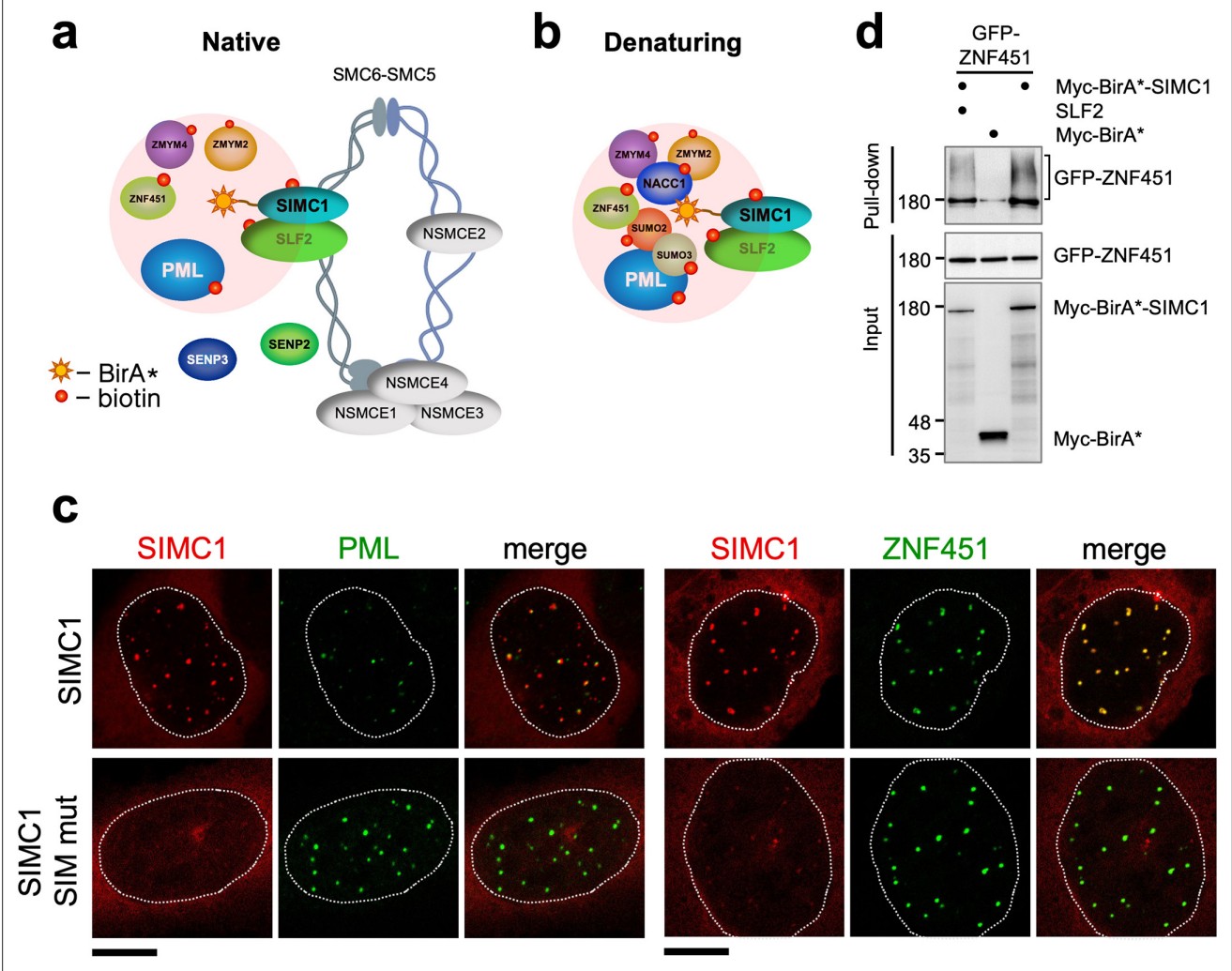

**Figure 3.** SIMC1 interacts with SMC5/6 and SUMO pathway factors. (**a, b**) A subset of SIMC1 interacting proteins identified by BioID screen carried out under native (**a**) or denaturing (**b**) lysis conditions, respectively. Pink circle illustrates the biotinylation radius of BirA*. BioID datasets are available in the PRIDE database (https://www.ebi.ac.uk/pride/archive) under the accession code PRIDE PXD033923 and in *Supplementary file 2*. (**c**) Representative immunofluorescence images of U2OS cells overexpressing mCherry-SIMC1 or mCherry-SIMC1 SIM mut (mutations: FIDL to AADA (26–29 aa) and VIDL to AADA (45–48 aa)) that were stained for PML (left three panels); or were co-expressing GFP-ZNF451 (right three panels). The cells were stained with DAPI to mark nucleus (dotted lines). Scale bar 10 µm. (**d**) HEK293 cells overexpressing ZNF451, BirA*-SIMC1, SLF2, BirA* control in indicated combination were cultured in the presence of 50 µM supplemental biotin. The total cell extracts and streptavidin pulldown were analyzed by western blot using anti-GFP or Myc antibodies. Full and unedited blots provided in *Figure 3—source data 1*.

The online version of this article includes the following source data for figure 3:

**Source data 1.** Full and unedited blots corresponding to panel (**d**).

SMC5/6 complex subunits were recovered (*Figure 3b*). These results confirm our SMC5 BioID results (*Figure 1d*), and together, they establish SIMC1 as an Nse5-like SMC5/6 cofactor.

In both conditions, multiple SUMO pathway factors were also identified as SIMC1 proximal, including PML, SUMO2/3, and ZNF451 (*Figure 3a and b*, *Supplementary file 2* and PRIDE database: PXD033923), a SUMO2/3 specific E3 ligase that can drive SUMO2/3 chain formation (*Cappadocia et al., 2015*; *Eisenhardt et al., 2015*). Supporting this, SIMC1 was previously identified using a computational string search for proteins that contain multiple SIMs and was found to bind SUMO2 polymers in vitro via its tandem N-terminal SIMs (*Sun and Hunter, 2012*; *González-Prieto et al., 2021*). Interestingly, SMC5/6 localizes at PML bodies in cancer cells (e.g. U2OS) that use the alternative lengthening of telomeres (ALT) mechanism to maintain their telomeres (*Potts and Yu, 2007*). Thus, SIMC1 may recruit SMC5/6 to SUMO-rich PML bodies.

We further tested SIMC1's proximity to the BioID hits PML and ZNF451. Expressed SIMC1 forms nuclear foci that colocalize with both endogenous PML NBs and expressed ZNF451, in a manner dependent on the SIM motifs of SIMC1 (*Figure 3c*). Moreover, epitope-tagged ZNF451 co-purifies with Myc-BirA*-SIMC1 but not a Myc-BirA* control (*Figure 3d*). SIMC1's proximity to ZNF451 is not enhanced by co-expression with SLF2, consistent with SIMC1 having direct SIM-mediated contacts with both SUMO and SUMO pathway factors (*Figure 3d*). These data validate the interface between SIMC1 and the SUMO pathway, and link SMC5/6 and SUMO-regulated PML NBs via SIMC1.

## SIMC1 recruits SMC5/6 to SV40 replication centers

Because SV40 has been reported to localize to PML NBs (*Ishov and Maul, 1996*), SIMC1 may recruit SMC5/6 to SV40 viral replication centers. To test this, we first confirmed the colocalization of SMC5/6, PML, and SV40 LT in permissive HEK293 cells transfected with replication competent SV40. As anticipated, SV40 replication centers (as marked by LT) contain endogenous SMC5/6 and often colocalize with PML NBs (*Figure 4a*). Thus, SMC5/6 is a novel DNA repair and replication factor found at sites of SV40 replication. We then generated SIMC1 knockout HEK293 cells, which are viable and show no overt changes to cell cycle distribution or signs of spontaneous DNA damage, as determined by FACS analysis and comet assay, respectively (*Figure 4—figure supplement 1*).

In SIMC1 null cells, SMC5/6 localization to SV40 replication centers is abolished (*Figure 4a*). Re-expression of SIMC1 restored SMC5/6 colocalization with SV40 LT and SIMC1 (*Figure 4b and c*). Despite similar expression levels as wild-type, SIMC1 with mutated SIMs only partially restored SMC5/6 colocalization with SV40 LT, indicating that it is hypomorphic (*Figure 4b–d*). In addition, the SIM mutant of SIMC1 failed to colocalize with PML and ZNF451 NBs (*Figure 3c*). Thus, SIMC1 and its interaction with SUMO are required for SMC5/6 localization to PyVRCs.

## SIMC1 and SLF2 form an Nse5/6-like structure

To gain insights into the molecular mechanism underlying SMC5/6 regulation, we analyzed the structures of the Nse5- and Nse6-like domains of SIMC1 and SLF2. AlphaFold structure prediction has provided highly reliable structures for proteomes (*Jumper et al., 2021*). The AlphaFold model of SIMC1 confirms that the N-terminus is disordered and reveals that the Nse5-like region forms an α-solenoid-like helical structure consisting of 17 α-helices (*Figure 5a*). *S. cerevisiae* Nse5 (ScNse5) also has an α-solenoid-like structure as determined previously by cryo-EM and X-ray crystallography *Taschner et al., 2021*; *Yu et al., 2021*; therefore, we compared SIMC1 and ScNse5 structures. Because automated superimposition did not work well, we used remote sequence homology. The HHpred matches are limited to the core region of SIMC1 ranging from α5 to α14 (*Figure 5—figure supplement 1*). Mapping the matched α-helices on the structures indicated that the matched regions are topologically similar (*Figure 5—figure supplement 2a and b*). Manual alignment of the matched regions overlaid the N-terminal region α1-α4 as well (*Figure 5—figure supplement 2c*). It also visualized additional helices (termed α3.1, α4.1, and α4.2) unique to ScNse5 in its N-terminus. By excluding these helices, the two structures could be superimposed with an r.m.s.d. of 7.8 Å between the structurally aligned 225 Cα atoms (*Figure 5—figure supplement 2d*). Similarly, an HHpred search using a ScNse5 sequence excluding the additional helices extended the match to include the α1-α4 region (*Figure 5—figure supplement 3*). While these results seemed to indicate that SIMC1's α15-α17 at its C-terminus was not conserved in yeast, an unpublished PDB entry (ID: 7SDE) of a cryo-EM structure of ScNse5/6 containing these additional helices has become available. Superimposing this Nse5 onto the SIMC1 model (r.m.s.d.=6.3 Å for 207 CA atoms) extended the overlaid region to include these C-terminal helices (*Figure 5a* and *Figure 5—figure supplement 4*). Thus, SIMC1's α-solenoid domain is mostly similar to Nse5.

AlphaFold reveals that the C-terminal Nse6-like region of SLF2 also adopts an α-solenoid-like structure. Although an HHpred search with SLF2 did not identify ScNse6 (*Figure 2—figure supplement 1*), automated superimposition aligned the SLF2 and ScNse6 structures well, with an r.m.s.d. of 5.7 Å between the structurally aligned 128 Cα atoms (*Figure 5b*). The overlay suggests that SLF2 and Nse6 structures are very similar except for a few α-helices (α1-α3) in the N-terminus of SLF2 that are missing in ScNse6 (*Figure 5b* and *Figure 5—figure supplement 5*). We noted that the SIMC1 and SLF2 α-solenoids are also similar; they can be superimposed well from their N- to C-termini (an r.m.s.d. of 5.6 Å for 247 Cα atoms, *Figure 5c*). However, the similarity between SIMC1 and SLF2 appears to be limited

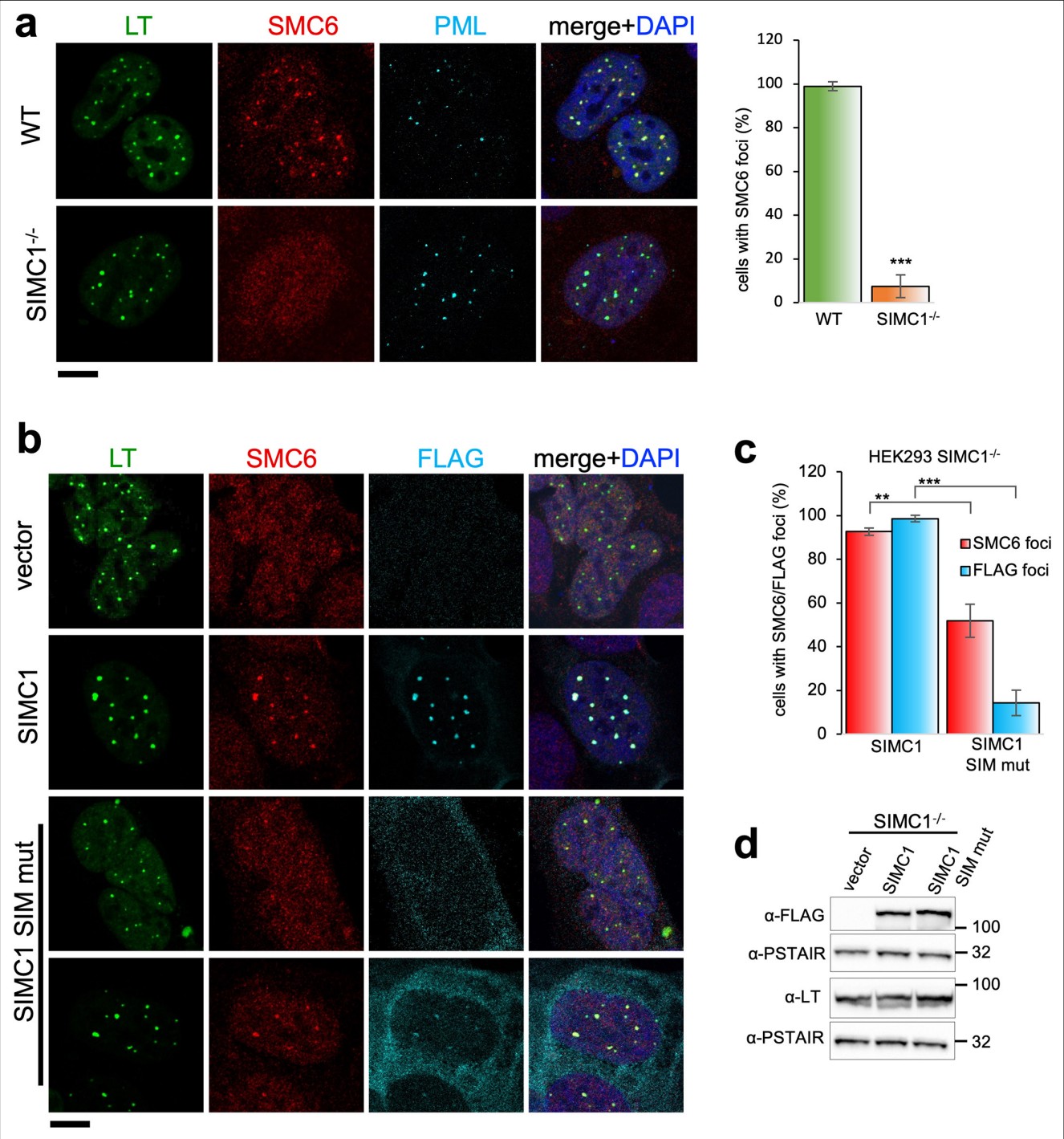

**Figure 4.** SIMC1 recruits SMC5/6 to SV40 replication centers. (**a**) Representative immunofluorescence images of SV40 LT (green), SMC6 (red) and PML (light blue) in HEK293 and HEK293 SIMC1$^{-/-}$ cells fixed 48 h after SV40 transfection. Scale bar 10 μm. The bar graph shows relative quantification of the number of cells containing SMC6 foci. A minimum of 175 cells with at least four SV40 LT foci were counted for each cell line. Primary data for graph in panel (**a**) provided in *Figure 4—source data 1*. (**b**) Representative immunofluorescence images of HEK293 SIMC1$^{-/-}$ cells with integrated empty vector or vectors expressing FLAG-SIMC1 or FLAG-SIMC1 SIM mut, respectively, 48 hr after SV40 transfection. SV40 LT (green), SMC6 (red), FLAG (light blue); Scale bar 10 μm. (**c**) Relative quantification of the number of cells containing SMC6 and FLAG foci (representative images shown in panel **b**). A minimum of 225 cells with at least four SV40 LT foci were counted for each cell line. Primary data provided in *Figure 4—source data 2*. (**d**) Immunoblot from HEK293 SIMC1$^{-/-}$ cells with integrated empty vector or vector expressing FLAG-SIMC1 or SIMC1 SIM mut, respectively, and transfected with a plasmid carrying SV40 genome. Cells were harvested 48 hr after transfection. PSTAIR serves as a loading control. Full and unedited blots provided in *Figure*

*Figure 4 continued on next page*

*Figure 4 continued*

**4—source data 3**. Statistics in (**a, c**): Means and error bars (s.d.) were derived from a minimum of three independent SV40 transfections representing biological replicates. (∗) p<0.05; (∗∗) p<0.005; (∗∗∗) p<0.0005; (n.s.) p>0.05 (two-tailed unpaired t-test).

The online version of this article includes the following source data and figure supplement(s) for figure 4:

**Source data 1.** Primary data for graphs in panel (**a**).

**Source data 2.** Primary data for graphs in panel (**c**).

**Source data 3.** Full and unedited blots corresponding to panel (**d**).

**Figure supplement 1.** Validation of HEK293 SIMC1-/- clone.

**Figure supplement 1—source data 1.** Primary data for graphs in panel (**b**).

**Figure supplement 1—source data 2.** Full and unedited blots corresponding to panel (**c**).

---

to topology (*Figure 5—figure supplement 6*); their sequences do not align, not even remotely, as HHpred fails to detect each other. ScNse5 and ScNse6 structures are too remote to superimpose.

Next, we attempted to predict the structure of the SIMC1-SLF2 complex using ColabFold and AlphaFold-Multimer (*Evans et al., 2021*; *Mirdita et al., 2021*). Both programs yielded the same structure with high confidence (*Figure 5—figure supplement 7*). To evaluate the accuracy of the predicted structures, we performed cryo-EM single-particle analysis on the SIMC1$^{284}$-SLF2$^{CTD}$ complex (*Figure 5—figure supplement 8* and *Table 1*). 3D reconstruction yielded an anisotropic and low-quality map, resolving side-chain densities of only bulky residues in the protein core (*Figure 5—figure supplement 9*). The limited orientations and alignability of the particles and the dynamic nature in the end regions of the α-solenoids together likely hampered the reconstruction. Nonetheless, the predicted SIMC1-SLF2 model could be docked into the map unambiguously and refined to ~3.9 Å resolution. The refined structure was strikingly similar to the predictions, with r.m.s.d.s of only ~1 Å for Cα atoms (*Figure 5—figure supplement 10*), validating the accuracy of the predicted structures.

The cryo-EM structure contains 425–858 aa of SIMC1 and 733–1158 aa of SLF2. SIMC1 and SLF2 α-solenoids face each other in a head-to-tail fashion through the concave surfaces of their slightly curved shapes (*Figure 5d*). Yeast Nse5 and Nse6 also interface in the same manner (*Figure 5e*). Overall, the ellipsoid-shaped SIMC1-SLF2 structure resembles the Nse5/6 structure (*Figure 5f*). The two structures could be superimposed with an r.m.s.d of 7.9 Å for 473 Cα atoms. The superimposition reveals that the regions consisting of the N-termini of SLF2/Nse6 and the C-termini of SIMC1/Nse5 are diverged: Nse6 lacks the N-terminal helices corresponding SLF2's α1-α3 and the C-terminal helices (α16-α17) of Nse5 do not overlap with the corresponding helices of SIMC1. In fact, the α16-α17 region of Nse5 does not make contacts with the N-terminus (α4) of Nse6 and appears to be dynamic as indicated by its high B-factors compared to the rest of the complex (*Figure 5—figure supplement 11a*). Notably, these C-terminal regions were not clearly resolved in the initial cryo-EM structure of ScNse5/6 and the X-ray ScNse5/6 structure. By contrast, the C-terminus of SIMC1 seems to be stable as indicated by its B-factor being comparable to the rest of the structure (*Figure 5—figure supplement 11b*). In comparison, the structures of the opposite ends consisting of the N-terminal region of SIMC1/Nse5 and the C-terminal region of SLF2/Nse6 converge well, suggesting the structural preservation of this region (*Figure 5f*).

A notable feature of the SIMC1-SLF2 structure is that the most N-terminal region (425–467 aa) of SIMC1 preceding its α-solenoid adopts an extended conformation, and a short region (431–451 aa) within it adopts a loop-like conformation that inserts into a cleft between the curved SIMC1 and SLF2 α-solenoids (*Figure 5—figure supplement 12a*). The loop contributes directly to the formation of the SIMC1-SLF2 interface; without this loop, the α-solenoids' interface would bury a surface area of only ~1500 Å², but the complete interface including the loop buries ~2500 Å². Consistently, our domain mapping experiments showed that the SIMC1 construct (457–872 a.a.) containing the entire α-solenoid but lacking the loop binds SLF2 more weakly than constructs containing the loop (*Figure 2c*). For comparison, the Nse5/6 interface has an open cleft between the subunits (*Figure 5—figure supplement 12b*). However, with the interface burying ~1800 Å², Nse5 and Nse6 can form a stable complex (*Pebernard et al., 2006*; *Palecek et al., 2006*; *Taschner et al., 2021*; *Yu et al., 2021*).

The interaction surfaces of both SIMC1 and SLF2 are highly charged (*Figure 5—figure supplement 13a*). The charges are distributed such that electrostatic interactions strengthen the interface.

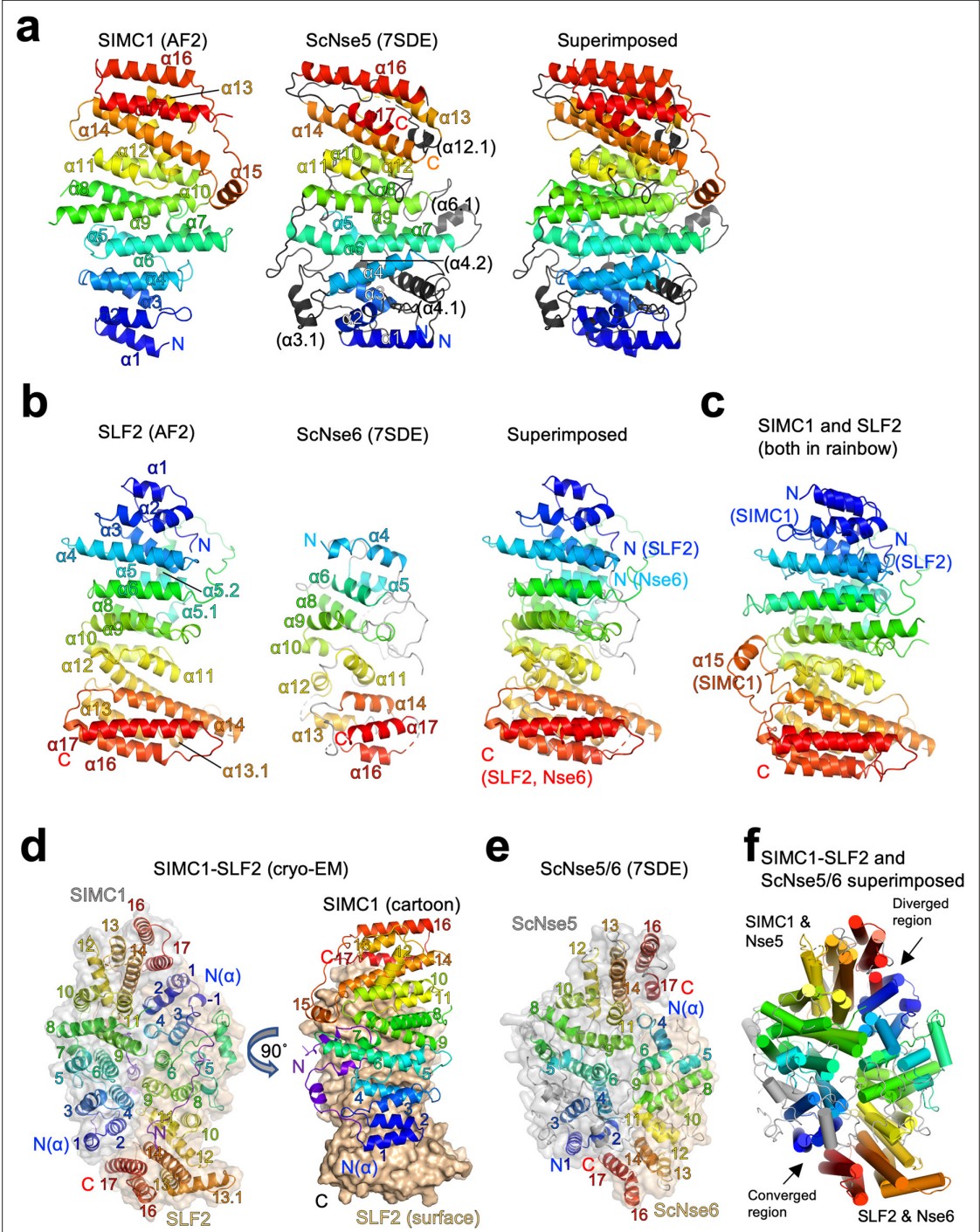

**Figure 5.** Structural analyses of the SIMC1-SLF2 complex. (**a**) The AlphaFold 2 (AF2) model of SIMC1 and its comparison to ScNse5. The C-terminal α-solenoid domain (468–858 aa) is shown on the left in the blue-red rainbow. The disordered 629–653 aa region between α8 and α9 is excluded for clarity. ScNse5 of the cryo-EM ScNse5/6 structure (PDB ID: 7SDE) is shown in the middle. Each α-helix is colored to match the corresponding α-helix of SIMC1. ScNse5's α-helices colored in black are not found in SIMC1. The superimposition of SIMC1 and ScNse5 is shown on the right. (**b**) AF2 model of SLF2 and its comparison to ScNse6. The C-terminal α-solenoid domain of SLF2 (755–1158 aa) is shown on the left. ScNse6 structure (PDB ID: 7SDE) shown in the middle is colored to match the α-helices of SIMC1. The superimposition of SLF2 and ScNse6 is shown on the right. (**c**) Overlay of SIMC1 and SLF2 AF2 models. (**d**) Cryo-EM structure of the SIMC1-SLF2 complex. Both SIMC1 and SLF2 structures contain a loop-like region at their N-termini, which are colored purple. The label "N(α)" indicates the N-terminus of each α-solenoid fold. On the right panel, SIMC1's surface is drawn only for its α-solenoid domain to clarify the N-terminal loop of SIMC1. (**e**) The cryo-EM structure of ScNse5/6 (PDB: 7SDE). (**f**) Superimposition of the SIMC1-SLF2 and ScNse5/6 complexes. Well-converged and diverged regions are indicated.

*Figure 5 continued on next page*

*Figure 5 continued*

The online version of this article includes the following source data and figure supplement(s) for figure 5:

**Figure supplement 1.** HHpred-aligned SIMC1 and ScNse5 (7LTO_A) sequences.

**Figure supplement 2.** Structural alignment of SIMC1 and ScNse5.

**Figure supplement 3.** The HHpred sequence alignment of an internally truncated ScNse5 and SIMC1.

**Figure supplement 4.** Structure-based sequence alignment of SIMC1 and ScNse5.

**Figure supplement 5.** Structure-based sequence alignment of SLF2 and ScNse6.

**Figure supplement 6.** Structure-based sequence alignment of SIMC1 and SLF2.

**Figure supplement 7.** Predictions of the SIMC1-SLF2 structure by ColabFold (**a**) and AlphaFold-Multimer (**b**).

**Figure supplement 8.** Single-particle cryo-EM data processing of the SIMC1-SLF2 complex.

**Figure supplement 9.** Cryo-EM density map of the SIMC1-SLF2 complex.

**Figure supplement 10.** Comparison of the predicted structures and the refined cryo-EM structure.

**Figure supplement 11.** The structures of ScNse5/6 (a, PDB: 7SDE) and the SIMC1-SLF2 complex (b) colored by B-factors.

**Figure supplement 12.** The N-terminal region to the α-solenoid of SIMC1.

**Figure supplement 13.** Properties of the SIMC1-SLF2 interface.

**Figure supplement 14.** Mutational analysis of the SIMC1-SLF2 interface.

**Figure supplement 14—source data 1.** Full and unedited blots corresponding to (**b**).

**Figure supplement 14—source data 2.** Full and unedited blots corresponding to (**c**).

Furthermore, the interface is enriched with conserved residues (*Figure 5—figure supplement 13b*), supporting the evolutionary conservation of the architecture from yeast to human.

We attempted to verify the SIMC1-SFL2 interface by mutational analysis. SIMC1 contacts SLF2 through the N- and C-terminal regions of its α-solenoid and the N-terminal loop structure that make additional contacts with the middle region of the SLF2 α-solenoid (*Figure 5—figure supplement 14a*). As the immunoprecipitation data of SIMC1 truncation constructs shown in *Figure 2c* already supported the contribution of the N-terminal loop structure in SLF2 binding, we focused on the residues on the α-solenoid. To test the N-terminal region, we simultaneously mutated four SLF2-interfacing residues on α4 to generate a construct named set A (*Figure 5—figure supplement 14a*). For the C-terminal region, we generated two constructs: set B with two mutations on α14 and set C with four mutations on α17. We introduced multiple mutations in each construct because disrupting such an extensive interface might require multiple perturbations. GFP-fused SIMC1 mutants were transiently expressed in HEK293 cells together with FLAG-tagged SLF2$^{CTD}$ and immuno-precipitated. As shown, the wild-type SIMC1 but not mutants were well detected in the precipitate (*Figure 5—figure supplement 14b*). Thus, we conclude that all the interface mutants tested disrupted the SIMC1 interaction.

We then set up reverse experiments with four SLF2 mutants (set A–D, *Figure 5—figure supplement 14a*). The set A residues on α1 and α2 in the N-terminal end of SLF2 contact the set B and C residues of SIMC1. The SLF2 set B residues at the N-terminal end of α4 contact α9 in the middle of SIMC1's α-solenoid. The SLF2 set C residues are also on α4 but contact the N-terminal loop of SIMC1. The SLF2 set D residues are on the turns preceding α9 and α12 and contact the SIMC1 set A residues. Myc-tagged SLF2$^{CTD}$ mutants were co-expressed with GFP-SIMC1 and precipitated using GFP-Trap. All mutants tested were significantly less detected in the precipitate compared to the wild-type, indicating reductions in the binding (*Figure 5—figure supplement 14c*). Thus, we conclude that our mutagenesis data support the structure of the SIMC1-SLF2 complex, which resembles the Nse5/6 structure.

## SIMC1 regulates SMC5/6 localization via its Nse5-like domain

To explore functional sites of the SIMC1-SLF2 complex, we returned to conservation analysis and found a composite patch consisting of conserved residues of the N-terminus of SIMC1 and the C-terminus of SLF2 (*Figure 6a*). As described above, this end of the complex is structurally preserved between yeast and human complexes (*Figure 5f*). Therefore, this patch may play a role in SMC5/6 regulation, possibly by binding to SMC5/6.

**Table 1.** Cryo-EM data collection/processing and model refinement statistics.

| Sample | Human SIMC1 (284-872)-SLF2 (635–1173) (EMD-25706, PDB 7T5P) | |
| --- | --- | --- |
| **Data collection and processing** | | |
| Condition | 0.4 mg/ml protein | 0.8 mg/ml protein with 0.128 mM DDM |
| Magnification | 73,000 × | 73,000 × |
| Voltage (kV) | 200 | 200 |
| Electron exposure (e⁻/Å²) | 66.8 | 66.9 |
| Number of frames per image | 46 | 46 |
| Defocus range (μm) | –0.5 to –1.8 | –0.5 to –1.8 |
| Pixel size (Å) | 0.567 | 0.567 |
| Symmetry imposed | C1 | C1 |
| Initial particle images (no.) | 1,873,621 | 1,065,073 |
| Final particle images (no.) | 31,383 | 8,443 |
| Map resolution (Å) FSC threshold | 3.4 0.143 | |
| **Refinement** | | |
| Initial model used | AlphaFold model | |
| Map resolution (Å²) FSC threshold | 3.9 0.5 | |
| Map sharpening B factor (Å²) | – | |
| Model composition | | |
| Chains | 2 | |
| Atoms | 13597 (Hydrogens: 6866) | |
| Protein residues | 822 | |
| Ligands | 0 | |
| B-factors (Å²) min/max/mean | | |
| Protein | 99.26/213.18/135.62 | |
| Ligand | – | |
| R.m.s. deviations | | |
| Bond lengths (Å²) | 0.003 | |
| Bond angles (°) | 0.591 | |
| Validation | | |
| MolProbity score | 2.00 | |
| Clashscore | 15.16 | |
| Rotamer outliers (%) | 0.26 | |
| Ramachandran plot | | |
| Favored (%) | 95.43 | |
| Allowed (%) | 4.57 | |
| Disallowed (%) | 0.00 | |

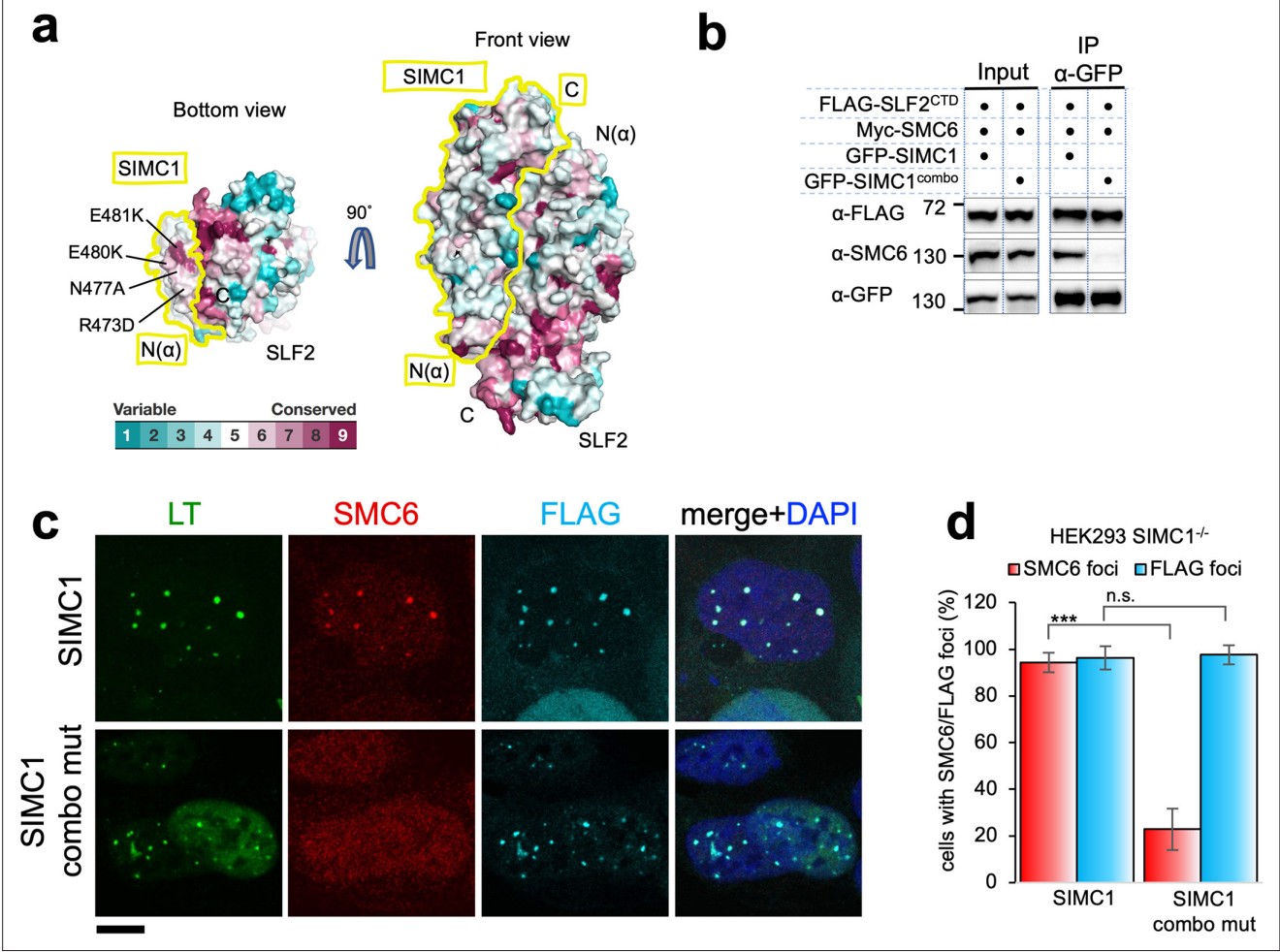

**Figure 6.** SIMC1 Nse5-like domain regulates SMC5/6 localization at PyVRCs. (**a**) Conservation mapping on the surface of the SIMC1-SLF2 complex. The conservation score obtained from Consurf server are shown by the color graduation as indicated. The boundary of SIMC1's surface is indicated by yellow lines for clarity. Mutated amino acids (R473D/N477A/E480K/E481K) in the α1 of SIMC1 α-solenoid are labeled. (**b**) Western blot of GFP-trap immunoprecipitation from HEK293 cells transfected with either GFP-SIMC1 or GFP-SIMC1 combo mutant in combination with FLAG-SLF2$^{CTD}$ and Myc-SMC6. Full and unedited blots provided in *Figure 6—source data 1* (**c**) Representative immunofluorescence images of HEK293 SIMC1$^{-/-}$ cells with integrated vectors expressing FLAG-SIMC1 or FLAG-SIMC1 combo mut, respectively. Cells were fixed 48 hr after SV40 transfection and stained with SV40 LT (green), SMC6 (red), FLAG (light blue) antibodies. Scale bar 10 μm. (**d**) Relative quantification of the number of cells containing SMC6 and FLAG foci (representative images shown in panel **c**). A minimum of 210 cells with at least four SV40 LT foci were counted for each cell line. Means and error bars (s.d.) were derived from three independent SV40 transfections representing biological replicates. (∗) p<0.05; (∗∗) p<0.005; (∗∗∗) p<0.0005; (n.s.) p>0.05 (two-tailed unpaired t-test). Primary data provided in *Figure 6—source data 2*.

The online version of this article includes the following source data for figure 6:

**Source data 1.** Full and unedited blots corresponding to panel (**b**).

**Source data 2.** Primary data for graphs in panel (**d**).

To test this, we took advantage of our SIMC1 null cells and expressed an epitope-tagged SIMC1 'combo' mutant that contains four mutations (R473D, N477A, E480K, and E481K), three of which are charge reversals in the α1 of SIMC1 α-solenoid (*Figure 6a*). We reasoned that these multiple and likely severe mutations would effectively disrupt the hypothesized interaction with SMC5/6. The mutated residues are all exposed to solvent and do not contact SLF2. Consistent with this, the SIMC1 combo mutant retained SLF2 binding, as demonstrated by co-immunoprecipitation (*Figure 6b*), contrasting the results with the SIMC1-SLF2 interface mutants that also had multiple mutations (*Figure 5—figure supplement 14*). The SIMC1 combo mutant expressed at a similar level to wild-type SIMC1 and colocalized with LT (*Figure 6b and c*). However, both the interaction of SIMC1 with SMC6 and the LT localization of SMC5/6 were markedly reduced with the combo mutant compared to wild-type SIMC1

(*Figure 6b, c and d*). Thus, these data indicate that the Nse5-like domain of SIMC1 is responsible for SMC5/6 recruitment but not for LT localization.

## SLF1 interacts with SLF2 through its split Nse5-like domain

The sequence of SLF1, the putative human orthologue of yeast Nse5, lacks detectable Nse5-like regions but contains an ankyrin repeat domain (ARD) (*Räschle et al., 2015*; *Adamus et al., 2020*). We turned to the AlphaFold model of SLF1, which reveals a flexible architecture containing BRCT domains at the N-terminus and the ARD and an α-solenoid-like helical domain in the C-terminus (SLF1$^{410}$; 410–1058 aa, *Figure 7a*). Notably, the ARD (726–935 aa) is an insert of the α-solenoid, which strikingly resembles SIMC1's Nse5-like domain, with an r.m.s.d. of 5.9 Å between the structurally aligned 300 Cα atoms (*Figure 7b*). We then ran an HHpred search with a C-terminal SLF1 sequence lacking the ARD (SLF1$^{410\_ΔARD}$), which identified Nse5 and SNI1 (*Figure 7—figure supplement 1a*). Thus, SLF1 does contain an Nse5-like domain split by an ARD in its sequence.

We then asked ColabFold for an SLF1$^{410}$-SLF2 structure, which predicted with high confidence a model similar to the SIMC1-SLF2 structure (*Figure 7c*, *Figure 7—figure supplement 2a*). Because the ARD does not contact SLF2, we repeated the prediction with SLF1$^{410\_ΔARD}$, which produced the same structure (*Figure 7—figure supplement 2b*). Predictions using a short SLF1 lacking the C-terminal half of the split Nse5-like domain (SLF1$^{410\_ΔC-half}$; 410–935 aa) and another SLF1 containing only the N-terminal half of the split Nse5-like domain (SLF1$^{410\_N-half}$; 410–725 aa) failed to assemble a reliable complex (*Figure 7—figure supplement 2c, d*). These results suggest that the complete Nse5-like domain is necessary and sufficient for the association of SLF1 with SLF2.

We then experimentally verified that expressed SLF1$^{410}$ binds to the SLF2$^{CTD}$ (*Figure 7d*). Consistent with the in silico results, SLF1$^{410\_ΔARD}$ retained interaction with SLF2$^{CTD}$, but SLF1$^{410\_ΔC-half}$ did not do so (*Figure 7d*). Next, we asked if SIMC1 and SLF1 form exclusive complexes with SLF2. SIMC1, SLF1, and SLF2 were co-expressed in cells and subjected to immunoprecipitation of either SLF1 or SLF2. Whilst both SIMC1 and SLF1 are found in SLF2 immunoprecipitates, only SLF2 was detected following immunoprecipitation of SLF1 (*Figure 7e*). Therefore, either SLF1 or SIMC1 forms an Nse5/6-like complex with SLF2, which agrees with the structural data showing that SLF1 and SIMC1 bind to the same surface on SLF2. The mutually exclusive binding of SIMC1 and SLF1 to SLF2 is further supported by recent proteome-wide mass spectrometry analyses (*Figure 7—figure supplement 3*, *Dupont et al., 2021*; *Huttlin et al., 2021*).

We wondered if the structural and functional similarities between SIMC1 and SLF1 are encoded in the primary sequence. To test this, we ran an HHpred search with the SLF1 sequence against the human proteome, which identified SIMC1 (*Figure 7—figure supplement 1b*). To validate the HHpred result, we manually generated a sequence alignment between SIMC1 and SLF1 based on the structural alignment described above (*Figure 7—figure supplement 4*) and compared it with the HHpred alignment (*Figure 7—figure supplement 5*). While the registrations are slightly off from the structure-based alignments, the HHpred alignment matches most of the structurally corresponding helices, covering from α1 to α14, which is the first helix of the C-terminal half of the split Nse5 domain in the SLF1 sequence. Altogether, our results establish structural and functional (SLF2 binding) similarities between SLF1 and SIMC1. We conclude that the two proteins are distant paralogues.In the analysis described above, we focused on the α-solenoid domain to define SLF1 as a human orthologue of yeast Nse5. In the AlphaFold model of SLF1, the α-solenoid domain is preceded by several α-helices whose confidence levels are not very high (*Figure 7—figure supplement 6a*). AlphaFold Multimer prediction using this longer SLF1 construct including the N-terminal helices (SLF1$^{336}$: 336–1058 aa) and SLF2$^{CTD}$ yielded overall the same structure as the SLF1$^{410}$-SLF2$^{CTD}$ structure, except the N-terminal extension (*Figure 7—figure supplement 6b*). SLF1's N-terminal α-helix remains at a similar position on the α-solenoid of the original AlphaFold model and interfaces with SLF2 (*Figure 7—figure supplement 6c*). Like SIMC1, SLF1's SLF2 interacting surface including the N-terminal α-helix is charged (*Figure 7—figure supplement 7a*). It also contains conserved residues, albeit less conserved than SIMC1 (*Figure 7—figure supplement 7b*). Thus, SLF1 sequences may be more diverged than SIMC1.

We attempted to verify the predicted structure using SLF1 and SLF2 variants (*Figure 7—figure supplement 8a*). Myc-tagged SLF2$^{CTD}$ precipitated more SLF1$^{336}$ than SLF1$^{410}$, supporting the participation of the N-terminal region in the SLF1-SLF2 interaction (*Figure 7—figure supplement 8b*). Thus, while structurally different, the N-terminal regions of SLF1 and SIMC1 α-solenoids are both involved

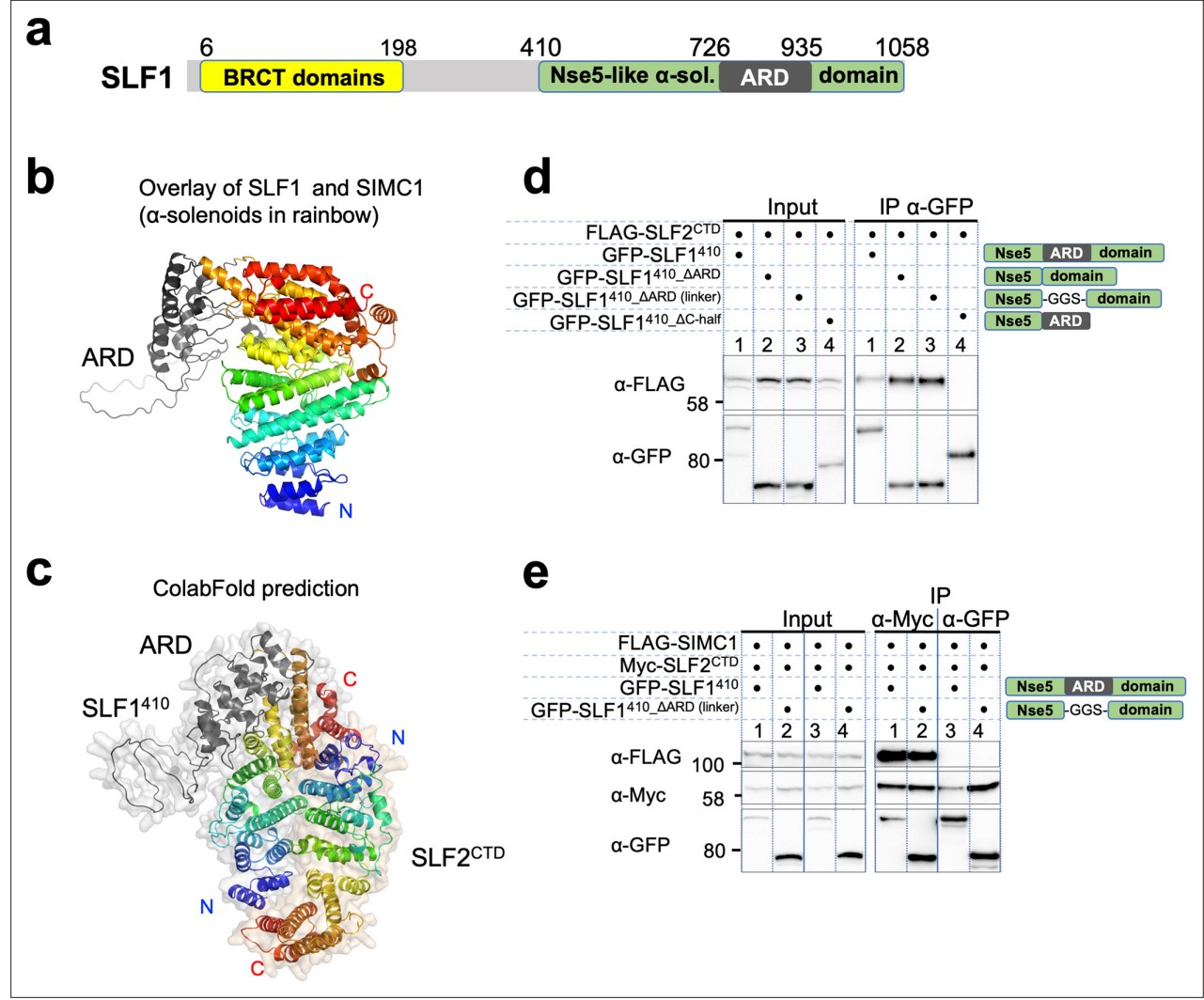

**Figure 7.** SLF1 interacts with SLF2 through its split Nse5-like domain. (**a**) Schematic of SLF1 domain boundaries on the AlphaFold model. (**b**) Superposition of the α-solenoid domains of SLF1 and SIMC1. (**c**) The ColabFold-predicted structural model of the SLF1$^{410}$-SLF2$^{CTD}$ complex. (**d**) Western blot of GFP-Trap immunoprecipitation from HEK293 cells transiently transfected with plasmids expressing the respective protein combination. Schematic on the right represents domains in SLF1 truncated variants. Full and unedited blots provided in *Figure 7—source data 1* (**e**) Western blot of Myc-Trap or GFP-Trap immunoprecipitation from HEK293 cells transfected with FLAG-SIMC1, Myc-SLF2$^{CTD}$ and GFP-SLF1$^{410}$ or GFP-SLF1$^{410\_\Delta ARD \,(linker)}$, respectively. Domains of SLF1 constructs are shown in the scheme on the right side. Full and unedited blots provided in *Figure 7—source data 2*.

The online version of this article includes the following source data and figure supplement(s) for figure 7:

**Source data 1.** Full and unedited blots corresponding to panel (**d**).

**Source data 2.** Full and unedited blots corresponding to panel (**e**).

**Figure supplement 1.** Results of an HHpred search with SLF1$^{410\_\Delta ARD}$ (410–725 aa +935–1058 aa) against yeast and plant sequences (**a**) and human sequences (**b**).

**Figure supplement 2.** ColabFold structure prediction with SLF1 variants and SLF2$^{CTD}$ (755–1159 aa).

**Figure supplement 3.** BioPlex interactome nodes for SLF2, SIMC1, and SLF1.

**Figure supplement 4.** Structure-based sequence alignment of SIMC1 and SLF1.

**Figure supplement 5.** HHpred-aligned SLF1 and SIMC1 sequences.

**Figure supplement 6.** AlphaFold structural analysis of SLF1 containing the N-terminal region.

**Figure supplement 7.** Properties of the SLF1-SLF2 interface.

**Figure supplement 8.** Mutational analysis of the SLF1-SLF2 interface.

**Figure supplement 8—source data 1.** Full and unedited blots corresponding to (**b**).

*Figure 7 continued on next page*

*Figure 7 continued*

**Figure supplement 8—source data 2.** Full and unedited blots corresponding to (**c**).

**Figure supplement 8—source data 3.** Full and unedited blots corresponding to (**d**).

**Figure supplement 9.** Conservation analysis of the SLF1-SLF2 complex model.

in SLF2 binding. We then generated three SLF1[336] mutants (set A/B/C) with each containing multiple mutations. Like SIMC1, the mutations in set A/B/C were on α4, α14, and α17 (*Figure 7—figure supplement 8a*). The IP data show no interaction between all three mutants and SLF2[CTD], supporting the predicted structure (*Figure 7—figure supplement 8c*).

We also tested SLF1 binding of the same SLF2[CTD] mutants used for validating the SIMC1 interaction. As observed above in the SIMC1 binding experiments (*Figure 5—figure supplement 14c*), SLF2[CTD] set A and set D showed no binding to SLF1[336] (*Figure 7—figure supplement 8d*), supporting our conclusion that SLF2 uses the same surface to form mutually exclusive complexes with SIMC1 and SLF1. A potential caveat to this interpretation is the significantly elevated expression of these two SLF2 mutants, which could be the result of aggregation caused by protein destabilization. In contrast, when these two SLF2 mutants were co-expressed with SIMC1, both mutants (and SIMC1) were expressed much less than the wild-type proteins (*Figure 5—figure supplement 14c*). We favor the interpretation that the binding-incompetent SLF2 and SIMC1 proteins are prone to degradation. As such, currently, it is difficult to completely rationalize these observations, and addressing these issues would require additional mutational analyses using purified mutant proteins. Nonetheless, our original interpretation of the SLF2 set A and D mutant data are consistent with the competitive binding of SIMC1 and SLF1 to SLF2 (*Figure 7e*) and the similar structures of the SIMC1-SLF2 and SLF1/2 complexes (*Figures 5d and 7c*).

SLF2[CTD] set B and C are only slightly defective in SLF1 binding, contrasting the SIMC1 results. SLF2 set B (C820W, I821W) residues on α4 make contacts with α9 in the middle of SLF1's α-solenoid. These contacts are not shielded by other contacts and therefore the side-chains of the introduced tryptophan residues may fit the surrounding space, thereby not disrupting the interface. SLF2 set C residues (M831R and I835R), which are also on α4 and contact the N-terminal extension of SLF1, are in a similar structural situation as set B residues. Therefore, the incorporated arginine side-chains may be accommodated by the surrounding space without causing a major disruption in the interface. These residues are likely not able to fit in the interface to SIMC1, causing defective SIMC1 binding (*Figure 5—figure supplement 14c*). While the IP data provides only qualitative information for interactions, the differences observed with set B and set C mutants in SIMC1 and SLF1 binding suggest that SLF2 uses its residues differently to interact with the two binding partners with similar but distinct surface properties.

The structural similarity between the SIMC1-SLF2 and SLF1/2 complexes implies that the two complexes would share a mechanism for SMC5/6 regulation. The N-terminal helix of SLF1's α-solenoid seems to be somewhat less conserved than that of SIMC1 (*Figure 7—figure supplement 9*). Indeed, the four residues that were mutated in the SIMC1 combo mutant are not strictly the same. Nonetheless, there is a glutamic acid shared between SLF1 and SIMC1, and the other three residues are all hydrophilic residues in both proteins. Thus, SLF1 may also use these residues to regulate SMC5/6.

## SLF1 but not SIMC1 is recruited to sites of DNA damage

Having established both SIMC1 and SLF1 as Nse5-like proteins, we investigated potential functional overlap between the two proteins. Thus far, our data suggest separate roles for SIMC1 and SLF1 in the recruitment of SMC5/6 to sites of viral replication or DNA damage, respectively. To explore this possibility, we stably expressed epitope tagged SIMC1 or SLF1 in U2OS cells and tested the recruitment of each to laser induced DNA damage, as done previously for SLF1 (*Räschle et al., 2015*). As expected, SLF1 colocalizes at laser stripes with gamma-H2AX, a marker of DNA damage (*Figure 8a*). In contrast, SIMC1 does not similarly colocalize with gamma-H2AX (*Figure 8a*).

We also tested whether SLF1, like SIMC1, localizes at SV40 replication centers. Overexpressed SLF1 localized proximal to LT foci in a qualitatively different manner to SIMC1 in wild-type HEK293 cells (*Figure 8b*). Moreover, this SLF1 localization requires SIMC1, as it is greatly reduced in SIMC1 null cells (*Figure 8b and c*). Whereas SIMC1 appears to be within PML NBs and LT foci, consistent

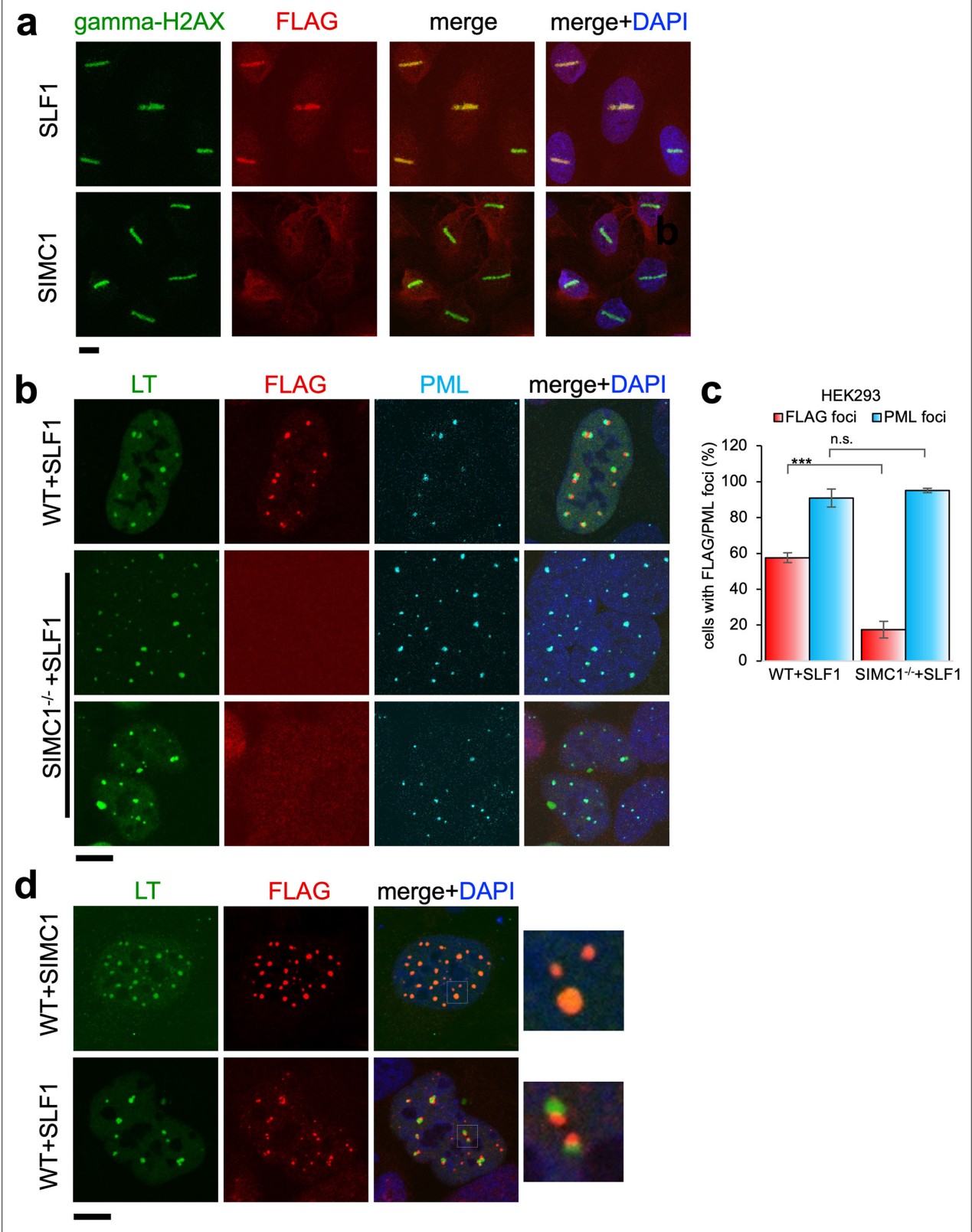

**Figure 8.** Recruitment of SLF1 but not SIMC1 to laser-induced DNA damage. (**a**) U2OS cells stably expressing either FLAG-SLF1 or FLAG-SIMC1 were exposed to laser microirradiation. Cells were fixed 1 hr later and stained with gamma-H2A.X (phosphorylation of histone H2A.X Ser139) and FLAG antibodies. Scale bar 10 µm. (**b**) Localization of SLF1 to LT foci depends on SIMC1. Representative immunofluorescence images of HEK293 WT and SIMC1-/- cells with integrated vector expressing FLAG-SLF1. Cells were fixed 48 hr after SV40 transfection and stained with SV40 LT (green), FLAG (red),

*Figure 8 continued on next page*

*Figure 8 continued*

PML (light blue) antibodies. Scale bar 10 µm. (**c**) Relative quantification of the number of cells containing FLAG and PML foci representative images shown in panel (**b**). A minimum of 267 cells with at least four SV40 LT foci were counted for each cell line. Means and error bars (s.d.) were derived from three independent SV40 transfections representing biological replicates. (∗) p<0.05; (∗∗) p<0.005; (∗∗∗) p<0.0005; (n.s.) p>0.05 (two-tailed unpaired t-test). Primary data provided in *Figure 8—source data 1*. (**d**) Comparison of localization of FLAG-SIMC1 and FLAG-SLF1 (red), stably expressed from integrated vectors, and LT foci (green) in HEK293 cells. Cells fixed 48 hr after SV40 transfection. Scale bar 10 µm.

The online version of this article includes the following source data and figure supplement(s) for figure 8:

**Source data 1.** Primary data for graph in (**c**).

**Figure supplement 1.** SIMC1 and SLF1 expression relative to beta-actin upon transfection of SV40 vector or control vector expressing GFP.

**Figure supplement 1—source data 1.** Primary data for graph in supplement.

with its SUMO binding properties, SLF1 appears at the periphery of LT foci (*Figure 8d*). Notably, we did not identify endogenous SLF1 in LT foci in our SMC5 BioID analysis, suggesting that the observed localization of overexpressed SLF1 may be an artifact.

Finally, we considered the possibility that SIMC1 expression is induced upon LT expression, providing a mechanism for the differential regulation of SMC5/6 by SIMC1 and SLF1. However, expression of both SIMC1 and SLF1 were similarly, and only marginally, induced by expression of LT (*Figure 8—figure supplement 1*). Therefore, transcriptional induction of SIMC1 by LT does not explain the different roles of SIMC1 and SLF1. Overall, these data support the independent/dominant roles of SIMC1 and SLF1 in viral challenge or DNA repair, respectively.

## Discussion

Our study identifies SIMC1 as an elusive human orthologue of yeast Nse5, establishes that SLF1 is also an Nse5 orthologue, and reveals evolutionary specialization of these key regulators of SMC5/6 function. SIMC1 and SLF1 form exclusive complexes with the Nse6 ortholog SLF2. Like Nse5/6, SIMC1-SLF2 and SLF1/2 function as SMC5/6 recruiters, and notably, they play distinct roles (*Figure 9*). We showed that SIMC1-SLF2 localizes SMC5/6 to PyVRCs, whereas SLF1/2 is known to recruit SMC5/6 to DNA lesions (*Räschle et al., 2015*). The different readers of post translational modifications (PTMs) in each protein likely mediate such distinct targeting.

Proteins use tandem SIMs to recognize SUMOylated proteins (*Sun and Hunter, 2012*) and enter SUMO-rich PML NBs via multivalent SIM-SUMO interactions (*Banani et al., 2016*). SIMC1 also relies on its SIMs to localize at PML NBs, thereby mediating the enrichment of SMC5/6 at PyVRCs. Once at viral replication centers, SIMC1, like yeast Nse5, may promote the SUMO ligase activity of SMC5/6

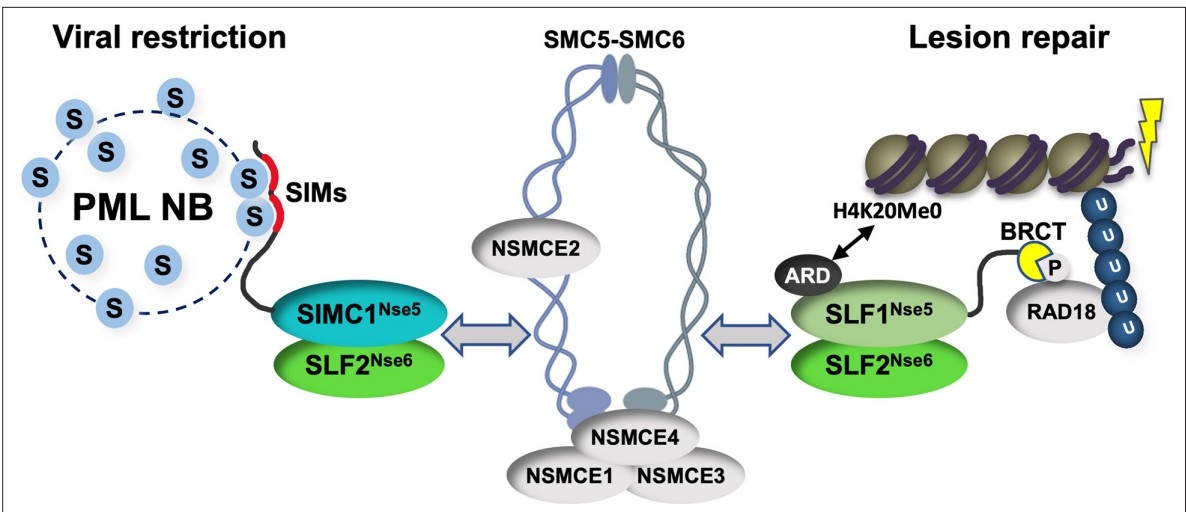

**Figure 9.** Schematic of SIMC1-SLF2 and SLF1/2 complex function. Human orthologues of yeast Nse5, SIMC1 and SLF1, form exclusive complexes via their C-terminal Nse5-like domains with the human Nse6 ortholog SLF2. The SIMC1-SLF2 complex localizes SMC5/6 to SV40 replication centers, whereas SLF1/2 targets it to DNA lesions, likely via recognition of different post translational modifications. S stands for SUMO; U for ubiquitin.

(*Oravcová et al., 2019b*; *Yu et al., 2021*; *Bustard et al., 2016*), which could in turn reinforce SIMC1-mediated SMC5/6 recruitment (*Figure 9*). Here, the interaction of SIMC1 with the poly-SUMO2 ligase ZNF451 could help generate SUMO2 chains (*Cappadocia et al., 2015*; *Eisenhardt et al., 2015*) that are in turn recognized by the SIM motifs of SIMC1.

SLF1 uses its N-terminal BRCT domains to bind to the DNA repair protein RAD18 at DNA lesions (*Räschle et al., 2015*). This mechanism parallels the recruitment of Smc5/6 to chromatin in yeast (*Oravcová et al., 2019b*; *Leung et al., 2011*). Brc1/Rtt107 uses its BRCT domains to bind Rad18 and gamma-H2AX at DNA lesions, while it also binds Nse6 within the Nse5/6 complex. Thus, by inclusion of BRCT domains, SLF1 appears to have combined Nse5 and Brc1/Rtt107 functions (*Figure 9*). SLF1 also uses its ARD to bind the unmethylated histone H4 tail (H4K20Me0), which is only found on nascent chromatin (*Nakamura et al., 2019*).

Complete understanding of the SMC5/6 recruitment mechanisms by SIMC1-SLF2 and SLF1/2 requires further work, but our data indicate that SIMC1's Nse5-like domain is involved. In addition, SLF2 has been reported to interact directly with SMC6 (*Adamus et al., 2020*). Thus, it is likely that SIMC1 or SLF1 and SLF2 together bridge the recruitment PTMs (e.g. SUMO, phospho-RAD18 and H4K20Me0) and the SMC5/6 core (*Figure 9*). Supporting this model, both Nse5 and Nse6 are essential for the recruitment of Smc5/6 to chromatin in yeast, and both contact Smc5/6 in crosslinking-MS studies (*Pebernard et al., 2006*; *Oravcová et al., 2019b*; *Hallett et al., 2021*; *Taschner et al., 2021*; *Yu et al., 2021*; *Bustard et al., 2012*). Nse5/6 also regulates Smc5/6 function by inhibiting its ATPase activity (*Hallett et al., 2021*; *Taschner et al., 2021*). Thus, the SIMC1-SLF2 and SLF1/2 complexes may act similarly to direct SMC5/6 activity.

Our discovery of SIMC1-dependent localization of SMC5/6 to PML NBs suggests a broad role of SMC5/6 in the antiviral response. Many viral genomes including HBV, HSV-1, HPV, SV40 and the pathogenic human polyomaviruses e.g., JCV are deposited next to, or seed the formation of, PML NBs (*Niu et al., 2017*; *Guion et al., 2019*; *Scherer and Stamminger, 2016*; *Hofmann et al., 2021*; *Everett, 2001*; *Gasparovic et al., 2009*). There are pro- and antiviral properties ascribed to PML NBs, but it is generally accepted that these bodies and some of the proteins they contain pose a barrier to productive infection by several viruses (*Niu et al., 2017*; *Guion et al., 2019*; *Scherer and Stamminger, 2016*; *Hofmann et al., 2021*; *Gasparovic et al., 2009*). In the case of HBV the restrictive roles of SMC5/6 and PML NBs are intertwined, as SMC5/6 transcriptionally silences HBV ccDNA only when localized at PML NBs (*Niu et al., 2017*). This is likely a more pervasive functional pairing given the localization of many viruses to PML NBs and the increasing number of these shown to be antagonized by SMC5/6 (*Decorsière et al., 2016*; *Xu et al., 2018*; *Gibson and Androphy, 2020*; *Dupont et al., 2021*).

In light of our discoveries, we propose that SIMC1 is a lynchpin of the antiviral functions of SMC5/6. Indeed, a recent CRISPR based genome-wide genetic study identified SIMC1, SLF2 and SMC5/6 components but not SLF1 as potent inhibitors of AAV (*Ngo and Puschnik, 2022*). This, together with the finding that SLF2 but not SLF1 is required for SMC5/6-mediated restriction of both HIV-1 and HBV (*Dupont et al., 2021*; *Abdul et al., 2022*), supports our proposal. Therefore, SIMC1-SLF2 may recruit SMC5/6-bound viruses to PML NBs using the SIM motifs in SIMC1, where viral transcription/replication is repressed. Further characterization of SIMC1 will thus be a promising strategy to establish the role of SMC5/6 in fighting pathogenic viruses and uncover the underlying molecular mechanisms.

# Materials and methods
## Construction of recombinant plasmids

Plasmid DNA was constructed either by standard restriction enzyme digestion and T4 ligase ligation-based cloning or using Takara In-Fusion HD Cloning Plus Kit. To make SIMC1 SIM mutations (FIDL to AADA and VIDL to AADA, corresponding to SIMC1 amino acids 26–29, and 45–48), a gBlock sequence containing all mutation was purchased (IDT). Combo, set mutations in SIMC1 Nse5-like domain and SLF2 set mutations were introduced in primers and gBlock sequences (IDT) and cloned into plasmids using In-Fusion kit (Takara). All plasmids created in this study have been verified by sequencing service provided by Eton Bioscience (San Diego). Plasmids used in this study are provided in Appendix 1 Key Resource table, additional details of plasmid construction are available upon request.

## Cell culture, transfection, stable line generation

Cells were cultured in Dulbecco's modified Eagle's medium (DMEM) supplemented with 10% fetal bovine serum (FBS), 1% antibiotic-antimycotic (Gibco) and maintained at 37 °C in humidified air with 5% $CO_2$. Human embryonic kidney cell lines HEK293T, human osteosarcoma cell line U2OS, and Phoenix ampho cells were obtained from Lazzerini-Denchi lab. HEK293 were provided by X. Wu lab. Flp-In–293 cell line was purchased from ThermoFisher (R75007). HEK293 and U2OS cell lines were authenticated by STR profiling (ATCC). Mycoplasma contamination was tested monthly by PCR detection (*Uphoff and Drexler, 2014*).

Transient plasmid transfections were generally carried out using TransIT-LT1 (Mirus) transfection reagent or Polyethylenemine (Polysciences Inc) at a 3:1 transfection reagent to DNA ratio. To generate stable cell lines with ectopic expression from plasmids, lentivirus or retrovirus was produced in HEK293T or Phoenix ampho cells, respectively, and used to infect target cell lines, followed by drug selection. Flp-In–293 cell lines were generated by co-transfecting 0.4 µg of the relevant FRT construct (Appendix 1 Key Resource table) and 3.6 µg of pOG44 (ThermoFisher) in 10 µl of Lipofectamine 2000 (ThermoFisher) diluted in 0.5 mL of OptiMEM (ThermoFisher), following the manufacturer's instructions. Hygromycin B was added 2 hr after transfection and selection continued until colonies started to appear.

A complete list of stable mammalian cell lines generated and reagents used in this study is provided (Appendix 1 Key Resource table).

## Cell line generation by CRISPR/Cas9

SIMC1 knockout clones were generated via CRISPR/Cas9 gene targeting by transient transfection of a hSpCas9 encoding plasmid (Addgene) and a pcDNA-H1 plasmid encoding specific sgRNA: guide1 (5´-gggtctgaacgacataacgc-3´), guide 2 (5´-cgcaggaaaaggactcgccc-3´), both in exon 5. To introduce stop codons by HR-mediated repair, a donor template with the STOP cassette sequence (GTCGGATC CTTTAAACCTTAATTAAGCTGTTGTAG) was used. The presence of the STOP cassette in clones from single cell were confirmed by sequencing, depletion of SIMC1 was verified by RT-qPCR and western blot. For RT-qPCR, total RNA was precipitated using RNeasy Plus Mini kit (Quiagen), cDNA synthetized by SuperScript III First-Strand Synthesis System for RT-PCR (Invitrogen) and SensiFAST SYBR No-ROX kit (Meridian Bioscience) was used for qPCR. Oligonucleotides are listed in the Appendix 1 Key Resource table.

## SILAC SMC5 BioID labeling and affinity purification of biotinylated proteins

For stable isotope labeling by amino acids in cell culture (SILAC), $8×10^6$ cells were seeded in a 15 cm plate with 45 mL of DMEM for SILAC (ThermoFisher) supplemented with foetal calf serum (10% v/v), 2 mM L-glutamine, 100 U/mL penicillinstreptomycin and the relevant amino acids: arginine-0/lysine-0 (light, 84 and 146 mg/L, Sigma-Aldrich, A6969 and L8662), arginine-6/lysine-8 (heavy, Cambridge Isotope Laboratories, CNLM-539-H-1 and CNLM-291-H-1). After 48 hr, biotin (stock: 50 mM in DMSO) was added to the medium to a final concentration of 50 µM for BirA*-SMC5 or 0.0125 µM for BirA*, and cells were cultured in the presence of biotin for 24 hr. Cells were detached by trypsinization, then counted and diluted to ~$1.25 × 10^8$ in 50 mL of PBS. Equal volumes of light and heavy-labelled cells were mixed as follows: 25 mL of BirA*-light with 25 mL of BirA*-SMC5-heavy and 25 mL of BirA*-heavy with 25 mL BirA*-SMC5-light. Cells were centrifuged at 300 *g* for 5 min, then incubated on ice for 15 min in 30 mL of 10 mM HEPES pH 7.6, 25 mM NaCl, 1.5 mM $MgCl_2$, 0.34 M sucrose, 10% v/v glycerol, 1×cOmplete protease inhibitors, 1 mM DTT (buffer A), supplemented with 0.1% v/v Triton X-100. The released nuclei were harvested by centrifugation at 1500 *g* for 5 min, washed once in 30 mL of buffer A supplemented with 0.1% v/v Triton X-100 and twice more in 30 mL of buffer A without Triton X-100. Purified nuclei were suspended in 2 mL of 1% v/v SDS, 10 mM EDTA pH 8, 20 mM HEPES pH 7.6, 1×cOmplete protease inhibitors (SDS buffer) and boiled for 10 min. Nuclear lysate was sonicated using a Branson 450 Sonifer equipped with a cone micro tip in five pulses of 15–20 s at minimum power settings to reduce sample viscosity. Nuclear lysate was diluted with 8 mL of 25 mM HEPES Ph 7.6, 150 mM NaCl, 1 mM EDTA pH 8, 1×cOmplete protease inhibitors, and 1 mM DTT (dilution/wash buffer), and incubated overnight at 4 °C with 400 µl of MyOne Streptavidin T1 Dynabeads that was pre-washed in dilution buffer. The beads were washed three times with 10 mL of dilution/wash buffer,

and eluted twice with 20 µl of 1×NuPAGE LDS Sample Buffer supplemented with 100 mM of DTT and 1 mM of biotin for 5 min at 95 °C.

## Sample preparation and mass spectrometry of SMC5 BioID with SILAC

The immunoprecipitated samples were run on the 10% SDS-PAGE for 10 min at 180 V to allow the proteins to move into the resolving gel. The gel was then fixed for 15 min in 7% acetic acid and 40% methanol solution and stained for 15 min with 0.25% Coomassie Blue G-250, 7% acetic acid, and 45% ethanol. After rinsing with deionized water to remove excess dye, In-gel digestion was performed in principle as described previously (*Shevchenko et al., 2006*) and detailed as follows. Each gel lane was cut separately using the new sharp scalpel, then minced, and transferred to an Eppendorf tube. Gel pieces were destained with 50% ethanol and 25 mM $NH_4HCO_3$ for 15 min to remove Coomassie dye. After removing supernatant, the gel pieces were dehydrated with 100% acetonitrile for 10 min on the rotator. Acetonitrile was discarded, and the samples were dried to completion using a vacuum evaporator (Eppendorf). The dried samples were then rehydrated and disulphide bonds in the proteins were reduced using 10 mM DTT in 50 mM $NH_4HCO_3$ pH 8.0 (reduction buffer) for 1 h at 56 °C. The buffer was removed, and cysteine residues of proteins were subsequently alkylated with 50 mM iodoacetamide and 50 mM $NH_4HCO_3$ pH 8.0 for 45 min at room temperature in dark. Samples were dehydrated again with 100% acetonitrile, then dried by vacuum evaporation. The fully dried gel slices were incubated with 1 µg trypsin per tube in 50 mM ammonium bicarbonate buffer at 37 °C overnight on a ThermoMixer (Eppendorf). Digested peptides were extracted twice with 150 µl of 30% acetonitrile, then with 150 µl of 100% acetonitrile to the gel pieces for 15 min at 25 °C while being agitated at 1400 rpm in a ThermoMixer (Eppendorf). Extracted peptides combined. The reductive dimethylation step was performed as described previously (*Boersema et al., 2009*). The labeled samples (each sample pair including a replicate were switched with the SILAC (Lys8/Arg10) labels) were mixed, and purified and desalted with C18 Stage-Tips (M3 company) as described (*Rappsilber et al., 2007*). The eluted peptides were loaded on the silica column of 75 µm inner diameter (New Objective) packed to 25 cm length with 1.9 µm, C18 Reprosil beads (Dr. Maisch Phases) using an Easy-nLC1000 Liquid Chromatography system (Thermo Scientific).

Peptides were separated on the C18 column using an Easy-nLC1000 Liquid Chromatography system (Thermo Scientific) with the following 2 h reversed-phase chromatography gradient: 0–4 min, 2–5% solvent B; 4–67 min, 5–22% solvent B; 67–88 min, 22–40% solvent B; 88–92 min, 40–95% solvent B; 92–97 min, 95% solvent B; 97–101 min, 95–2% solvent B; and 101–105 min, 2% solvent B (solvent B: 80% acetonitrile containing 0.1% formic acid) and directly sprayed into a Q-Exactive Plus mass spectrometer (Thermo Scientific) for the data acquisition. The mass spectrometer was operated in the positive ion scan mode with a full scan resolution of 70,000; AGC target $3x10^6$ max. IT = 20ms; Scan range 300–1650 m/z with a top10 MS/MS DDA method. Normalized collision energy was set to 25 and MS/MS scan mode operated at a resolution of 17,000; AGC target $1x10^5$ and max IT of 120ms.

## Nano-Trap immunoprecipitation

GFP and Myc-labeled proteins were immunoprecipitated using the Nano-Trap magnetic agarose (Chromotek) following the manufacturer's instructions. Briefly, about $10x10^6$ cells were harvested by trypsinization 48–72 h after plasmids transfection, lysed by incubation in 200 µl of dilution buffer (10 mM Tris pH 7.5, 150 mM NaCl, 0.5 mM EDTA) supplemented with 0.5% Nonidet NP40, 1 mM phenylmethylsulfonyl fluoride (PMSF), 1 x Halt Protease Inhibitor Cocktail (Thermo Fisher Scientific) for 30 min on ice with extensive pipetting every 10 min. Lysates were cleared by centrifugation at 17,000 *g* 12 min and input aliquot was incubated with 6 mM $MgCl_2$ and Benzonase (1:75, EMD Millipore). The remaining lysate was diluted with 300 µl of dilution buffer +supplements and incubated with 25 µl of GFP- or Myc-Trap magnetic agarose beads for 2 hr at 4 °C, followed by three washes in dilution buffer and elution into 2 x NuPAGE LDS Sample Buffer (ThermoFisher). Samples were analyzed in SDS-PAGE and western blot.

## SIMC1 BioID labeling and affinity purification of biotinylated proteins

BioID labeling was carried out essentially as described (*Nie et al., 2021*). Briefly, cells were cultured in media containing 50 µM biotin (Sigma) for 24 hr. Denaturing cell lysis and streptavidin pulldown was performed as described previously (*Roux et al., 2012*; *Nie et al., 2021*). Native lysis purification

was carried out as detailed below. Cell pellet was suspended in native lysis buffer (20 mM Tris, pH 7.5, 150 mM NaCl, 1 mM EDTA, 0.5% NP-40, 1 mM DTT, 6 mM MgCl$_2$, 1 mM PMSF) supplemented with Halt Protease Inhibitor Cocktail (ThermoFisher). The cell lysate was incubated on ice for 30 min with 50 units of Benzonase (EMD Millipore), then centrifuged at 4 °C at 16,000 *g* for 10 min. Supernatant, diluted with 3 volumes of 10 mM Tris, pH 7.5, 150 mM NaCl (native dilution buffer), was incubated for 2 h with Dynabeads MyOne Streptavidin C1 beads (ThermoFisher, 20 µl of beads was used in small-scale analysis; 75 µl in large-scale preparation for mass spectrometry). The beads were then washed three times with 10 mM Tris, pH 7.5, 500 mM NaCl (native wash buffer). Following washes, beads were either eluted with 35 µl of 2 x NuPAGE LDS Sample Loading Buffer (ThermoFisher) with 100 mM DTT at 100 °C for 5 min for western blotting, or stored in 100 mL of 8 M urea, 100 mM Tris, pH 8.5, for mass spectrometry analysis.

## Protein identification by mass spectrometry

Protein samples were reduced and alkylated by sequential incubation with 5mM Tris (2-carboxyethyl) phosphine for 20 min at room temperature and 10 mM iodoacetamide reagent in the dark at room temperature for additional 20 min. Proteins were digested sequentially at 37 °C with lys-C for 4 hr followed by trypsin for 12hours. After quenching the digest by the addition of formic acid to 5% (v/v), peptides were desalted using Pierce C18 tips (Thermo Fisher Scientific), dried by vacuum centrifugation, and resuspended in 5% formic acid. Peptides were fractionated online using reversed phase chromatography on in-house packed C18 columns as described before (*Jami-Alahmadi et al., 2021*). The 140 min gradient of increasing acetonitrile was delivered using a Dionex Ultimate 3000 UHPLC system (Thermo Fisher Scientific). Peptides were electrosprayed into the mass spectrometer by the application of a distal *2.2 kV* spray voltage. MS/MS data were acquired using an Orbitrap Fusion Lumos mass spectrometer operating in Data-Dependent Acquisition (DDA) mode consisting of a full MS1 scan to identify peptide precursors that were subsequently targeted by MS2 scans (Resolution = 15,000) using high energy collision dissociation for the remainder of the 3 s cycle time. Data analysis was performed using the Integrated Proteomics bioinformatic pipeline 2 (Integrated Proteomics Applications, San Diego, CA). Database searching was performed using the ProLuCID algorithm against the EMBL Human reference proteome (UP000005640 9606). Peptides identifications were filtered using a 1% FDR as estimated using a decoy database. Proteins were considered present in a sample if they had two or more unique peptides mapping to them. Relative comparisons between samples to identify candidate SIMC1 interacting proteins was done using raw peptide spectral counts.

## Immunofluorescence and microscopy

Immunofluorescence was performed essentially as described (*Nie et al., 2021*). Briefly, U2OS cells grown on coverslips were fixed in 4% formaldehyde in PBS for 10 min, then permeabilized with PBS containing 0.2% Triton X-100 for 5 min at room temperature. The cells were incubated with antibodies diluted in a blocking solution of 5% Normal Goat Serum (BioLegend) in PBS. HEK293 cells were grown on coverslips coated in 0.1% gelatin (type A, MP bio), fixed in 4% formaldehyde in PBS for 15 min, permeabilized in Triton X-100 buffer (0.5% Triton X-100, 20 mM Hepes, 50 mM NaCl, 3 mM MgCl$_2$, 300 mM sucrose) for 2 min. Antibodies were diluted in PBS containing 0.2% cold water fish gelatin (Sigma) and 0.5% BSA.

To visualize DNA, fixed cells were stained with 0.1 µg/mL 4',6-diamidino-2-phenylindole dihydrochloride (DAPI, Sigma). Coverslips were mounted in ProLong Gold Antifade (Invitrogen) to glass slides. Antibodies used for immunofluorescence are listed (Appendix 1 Key Resource table).

Confocal images were acquired with LSM 780 or 880 confocal laser scanning microscope (Zeiss) equipped with 63 x/1.4 oil immersion high NA objective lens, using standard settings in Zen software. Images were processed using ImageJ software (NIH).

## Western blotting

Whole cell lysate was prepared by lysis cells in RIPA buffer (150 mM NaCl, 1% Triton X-100, 0.5% sodium deoxycholate, 0.1% SDS, 50 mM Tris, pH 8), agitated for 40 min in 4 °C and centrifuged at 16,000 *g* for 20 min. The whole cell lysate or pulldown sample was combined with NuPAGE LDS Sample Loading Buffer and 100 mM DTT, separated by SDS-PAGE and transferred to nitrocellulose. Immunoblotting was performed as described (*Nie et al., 2017*). Antibodies used for immunoblotting

are listed (Appendix 1 Key Resource table). Protein detection was carried out using ECL chemiluminescent substrates (genDEPOT, ThermoFisher) on a ChemiDoc XRS + Molecular Imager (BioRad) using ImageLab software (BioRad).

## Flow cytometry

HEK293 cells were treated with 100 ng/mL nocodazole in 0.1% DMSO for 10 hr to induce G2/M arrest. Cells were fixed with ice-cold 70% EtOH, washed in PBS containing 1% BSA and stained with 50 μg/mL propidium iodide in the presence of 250 μg/mL RNAse A for 30 min in 37°C and 4°C overnight. Flow cytometry analysis was performed on a NovoCyte 3000 (ACEA Biosciences) using NovoExpress software.

## Alkaline comet assay

Endogenous level of DNA damage in HEK293 cells was evaluated using the CometAssay kit (Trevigen) according to the manufacturer's instructions. Control cells were treated with 1 mM MMS (Sigma) for 1 h to induce DNA damage. Cells were harvested by trypsinization and 500 cells were mixed with LMAgarose and spread on the slide. Slides were dried 20 min in 4 °C, incubated 40 min in kit lysis solution in 4 °C and 20 min in unwinding solution (300 mM NaOH, 1 mM EDTA, pH 13) in room temperature. Electrophoresis was performed in the unwinding solution in 4 °C, 300 mA, 30 min. Slides were washed twice in water, once in 70% EtOH, dried 15 min in 37 °C and stained with SYBR Gold (Thermo) for 30 min. Images were taken using Zeiss Axio Imager MI with 20 x objective.

## DNA damage by laser microirradiation

Protocol adapted from *Tampere and Mortusewicz, 2016*. Briefly, U2OS cells were plated in a 35 mm μ-Grid dish (Ibidi) a day before microirradiation. Cells were sensitized by 1 μg/ml Hoechst 33342 (Invitrogen) for 10 min and transferred to the stage incubator of LSM880 Airyscan confocal laser scanning microscope (Zeiss). Using the bleaching mode in ZEN software, DNA damage was induced by irradiation of a 5-pixel wide region with a 405 nm diode laser set to 15% power, 1 iteration, zoom 1, averaging 1, pixel dwell time 0.27 μs (speed 7). Cells were fixed 1 hour after the irradiation and stained with anti-phospho-Histone H2A.X (Ser139) and anti-D tag antibodies following the immunofluorescence protocol.

## Recombinant protein expression and purification

Recombinant human SIMC1 and SLF2 proteins were expressed in baculovirus-infected Spodoptera frugiperda (Sf9) insect cells. The cDNA fragment encoding SIMC1[284] (284–872 aa) was placed after the Tev-protease cleavable GST tag in a modified pACEBac vector. The cDNA fragment encoding SLF2[CTD] (a.a. 635–1173) was cloned after the Tev-cleavable Histidine tag in a modified pACEBac. Baculoviruses were generated from these vectors using the published protocol (*Berger et al., 2004*). Sf9 cells were co-infected with the baculoviruses harboring GST-SIMC1[284] or His-SLF2[CTD] and harvested after 48–50 hr. The cells were suspended in the lysis buffer (20 mM Hepes pH 7.5, 150 mM NaCl, 0.5 mM TCEP) supplemented with protease inhibitor cocktail (Pierce) and lysed using C3 pressure homogenizer (Avestin). The lysate was clarified by centrifugation at 40,000 *g*. Imidazole (pH 7, final 20 mM) was added to the supernatant, which was then poured onto a Ni affinity column (Thermo Fisher Scientific). The Ni resin was washed with five column volume of the lysis buffer. Proteins were eluted with 300 mM Imidazole. The Ni eluate was immediately poured onto a glutathione sepharose column (GoldBio). After wash, an aliquot of Tev protease was added to the resin, which was then left at 4 °C overnight. Proteins were then eluted in the lysis buffer, concentrated, and injected a Superdex 200 size exclusion column (GE Healthcare). The protein contents of the fractions were analyzed by SDS-PAGE, which was stained with Coomassie blue.

## ColabFold and AlphaFold-Multimer structural modeling

ColabFold and AlphaFold-Multimer were run on Google Colab platform using default settings.

## Cryo-EM data collection and processing

Cryo-EM grids were prepared in a cold room. A 3 μl drop of the SIMC1[284]-SLF2[CTD] complex (0.4 mg/ml protein in 20 mM HEPES pH 7.5, 150 mM NaCl, 0.5 mM TCEP or 0.8 mg/ml protein in 20 mM HEPES

pH 7.5, 150 mM NaCl, 0.5 mM TCEP, 0.128 mM DDM) was applied to a plasma-cleaned UltraAu-Foil R1.2/1.3 300-mesh grids (Quantifoil). Immediately after removing excess liquid by a filter paper (Whatman No. 1), the grid was rapidly dropped into liquid ethane using a manual plunger. Grids were stored in liquid nitrogen tank until use.

Data were collected in two sessions, which were done in a same manner as follows. Grids were observed on Talos Arctica 200 KeV transmission electron microscope (Thermo Fisher Scientific) equipped with a K2 Summit direct detector (Gatan). The microscope was aligned as reported previously (*Herzik et al., 2017*). Images were recorded in an automated manner using the Leginon software (*Suloway et al., 2005*). Data were collected as movie files with a frame rate of 200ms in the counting mode. The acquisition parameters are described in *Table 1*. Particle motion correction was performed using the MotionCor2 software (*Zheng et al., 2017*) in the Appion data processing pipeline (*Lander et al., 2009*). The frame-aligned images were imported into Relion 4.1b (*Kimanius et al., 2021*) and CTF estimation was performed using gCTF (*Zhang, 2016*). The first session on the condition of 0.4 mg/ml protein concentration collected 3332 micrographs. The grids contained many well-frozen areas, which enabled an efficient data collection. A total of 1,873,621 particles were picked from the first data set using Laplacian-of-Gaussian filter. Particle images were extracted at 2.268 Å per pixel with a box size of 100 pixels and subjected to 2D classification. 2D classes showing secondary structures were selected and subjected to *ab initio* reconstruction in cryoSPARC (*Punjani et al., 2017*), which yielded a model that appears to be consistent with 2D class average images. The *ab initio* model was 3D classified in Relion with K=3 and T=4. The particles belonging to the highest resolution model (620,042 particles, 41.0% of the total input particles) were selected and 3D auto-refined in Relion. The auto-refined particles were re-centered and re-extracted at 1.134 Å per pixel with a box size of 200 pixels, then were subjected to 2D classification. 203,464 particles were selected and subjected to 3D auto-refinement. The refined particles were combined with the particles from the second data set as described below.

Processing of the first data set made it clear that the orientation of the particles was limited. To alleviate this issue, DDM (0.75×critical micelle concentration) was added to the sample and the concentration of the protein was increased. The images of this sample appeared to show more different views of the particles; however, the grid contained too many empty holes in areas with thin ice, which made it difficult to obtain a large number of good particles. As such, the second session collected 1553 micrographs. A total of 1,065,073 particles were picked using Laplacian-of-Gaussian filter, extracted at 2.268 Å per pixel with a box size of 100 pixels, and then subjected to 2D classification. 128,371 particles belonging to well-resolved 2D class average images were selected and 3D auto-refined. The refined particles were re-centered and re-extracted at 1.134 Å per pixel with a box size of 200 pixels, and then subjected to 3D auto-refinement. The refined particles were 3D classified with no alignment with K=2 and T=8. 30932 particles belonging the better-resolved class (24.2% of the total input particles) were selected and 3D auto-refined. The particles were then combined with the refined particles from the first data set, yielding a total of 234,396 particles. The combined particles were subjected to 3D auto-refinement, followed by 3D classification without alignment (K=2, T=12). A total of 39,826 particles belonging to the better-resolved map (17.0% of the total input) were 3D auto-refined. The refined particles were subjected to CTF refinement, but no improvement was observed. Local resolution variation was calculated in Relion. Three-dimensional FSC analyses were performed on the remote 3DFSC processing server (https://3dfsc.salk.edu/, *Tan et al., 2017*). The final map was sharpened by DeepEMhancer (*Sanchez-Garcia et al., 2021*) on the COSMIC2 server (https://cosmic-cryoem.org/, *Cianfrocco et al., 2017*).

## Model building, refinement, and structural analyses

For model building, a SIMC1-SLF2 model predicted by AlphaFold-Multimer was docked into the sharpened cryo-EM map using the Dock in map module of PHENIX (*Liebschner et al., 2019*). The model was trimmed and adjusted into the map using COOT (*Emsley et al., 2010*) and refined using the Real-space refinement module of PHENIX. The map-to-model FSC was obtained from PHENIX. The refined model was validated using MolProbity (*Chen et al., 2010*). The buried surface areas of the SIMC1-SLF2 (PDB ID: 7T5P) and Nse5/6 (PDB ID: 7LTO) complexes were obtained from 'Protein interfaces, surfaces and assemblies' service PISA at the European Bioinformatics Institute (http://www.ebi.ac.uk/pdbe/prot_int/pistart.html, *Krissinel and Henrick, 2007*). Electrostatic potential was calculated

using APBS (*Baker et al., 2001*). Conservation scores were obtained from Consurf server (https://consurf.tau.ac.il, *Ashkenazy et al., 2016*). Structure figures were generated using Pymol (https://pymol.org), Chimera (*Pettersen et al., 2004*) and ChimeraX (*Goddard et al., 2018*).

## Material Availability

All materials produced during this work are available upon written request, in keeping with the requirements of the journal, funding agencies, and The Scripps Research Institute.

## Acknowledgements

We thank Dr. James DeCaprio and Dr. Xiaohua Wu for the kind gift of SV40 plasmids. We thank Chinatsu Otomo for the generation of baculoviruses. We thank Charly Chahwan (Synthex, Inc) for sharing his early insights into Nse5 orthologues. Computational resource for Cryo-EM data processing was supported by NIH S10OD021634. Support of the IMB Proteomics Core Facility and use of IMB's Q-Exactive Plus mass spectrometer is also gratefully acknowledged.

## Additional information

### Funding

| Funder | Grant reference number | Author |
| --- | --- | --- |
| National Institute of General Medical Sciences | GM136273 | Michael N Boddy |
| National Institute of General Medical Sciences | GM089788 | James A Wohlschlegel |
| National Institute of General Medical Sciences | GM092740 | Takanori Otomo |
| Deutsche Forschungsgemeinschaft | 393547839 - SFB 1361 | Helle D Ulrich |
| Deutsche Forschungsgemeinschaft | sub-project 07 | Helle D Ulrich |

The funders had no role in study design, data collection and interpretation, or the decision to submit the work for publication.

### Author contributions

Martina Oravcová, Minghua Nie, Resources, Formal analysis, Validation, Investigation, Visualization, Methodology, Writing – review and editing; Nicola Zilio, Formal analysis, Investigation, Methodology; Shintaro Maeda, Resources, Formal analysis, Investigation, Methodology; Yasaman Jami-Alahmadi, Data curation, Formal analysis, Investigation, Methodology; Eros Lazzerini-Denchi, Conceptualization, Writing – review and editing; James A Wohlschlegel, Data curation, Formal analysis, Supervision, Funding acquisition, Investigation, Methodology; Helle D Ulrich, Formal analysis, Supervision, Funding acquisition, Investigation; Takanori Otomo, Conceptualization, Data curation, Formal analysis, Supervision, Funding acquisition, Validation, Investigation, Visualization, Methodology, Writing – original draft, Writing – review and editing; Michael N Boddy, Conceptualization, Data curation, Formal analysis, Supervision, Funding acquisition, Validation, Investigation, Visualization, Methodology, Writing – original draft, Project administration, Writing – review and editing

### Author ORCIDs

Martina Oravcová (iD) http://orcid.org/0000-0001-6063-2227
Takanori Otomo (iD) http://orcid.org/0000-0003-3589-238X
Michael N Boddy (iD) http://orcid.org/0000-0001-7618-4449

### Decision letter and Author response

Decision letter https://doi.org/10.7554/eLife.79676.sa1
Author response https://doi.org/10.7554/eLife.79676.sa2

## Additional files

### Supplementary files
• Supplementary file 1. Mass spectrometry identification of SMC5 interacting proteins labelled by SILAC and BioID.
• Supplementary file 2. Mass spectrometry identification of SIMC1 BioID.
• MDAR checklist

### Data availability
The SMC5 and SIMC1 BioID datasets have been deposited to the PRIDE database as follows: Protein interaction AP-MS data: PRIDE PXD033923. Cryo-EM density map and atomic coordinates of the SIMC1-SLF2 complex have been deposited to the Electron Microscopy Data Bank and Protein Data Bank, respectively, under accession codes EMD-25706 and PDB 7T5P.

The following datasets were generated:

| Author(s) | Year | Dataset title | Dataset URL | Database and Identifier |
|---|---|---|---|---|
| Maeda S, Oravcova M, Boddy MN, Otomo T | 2022 | Cryo-EM structure of human SIMC1-SLF2 complex | https://www.rcsb.org/structure/7T5P | RCSB Protein Data Bank, 7T5P |
| Maeda S, Oravcova M, Boddy MN, Otomo T | 2022 | Cryo-EM structure of human SIMC1-SLF2 complex | https://www.ebi.ac.uk/emdb/EMD-25706 | Electron Microscopy Data Bank, EMD-25706 |
| Jami-Alahmadi Y, Boddy MN | 2022 | The Human Nse5 Orthologue SIMC1 Localizes SMC5/6 to Polyomavirus Replication Centers | http://www.ebi.ac.uk/pride/archive/projects/PXD033923 | PRIDE, PXD033923 |

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

# Appendix 1

### Appendix 1—key resources table

| Reagent type (species) or resource | Designation | Source or reference | Identifiers | Additional information |
|---|---|---|---|---|
| cell line (*Homo-sapiens*) | HEK293 | other | | Cell line maintained in X. Wu lab |
| cell line (*Homo-sapiens*) | HEK293 SIMC1-/- | This paper | clone A11 | Derived from HEK293 by CRISPR/ Cas9 |
| cell line (*Homo-sapiens*) | HEK293T; U2OS; Phoenix ampho | other | | Cell lines maintained in E. Lazzerini-Denchi lab |
| cell line (*Homo-sapiens*) | Flp-In–293 | ThermoFisher | R75007 | |
| cell line (*Spodoptera frugiperda*) | Sf9 | Expression Systems | | |
| transfected construct (human) | pOG44 Flp-Recombinase Expression Vector | ThermoFisher | V600520 | |
| transfected construct (human) | pDEST-BirA*–3FLAG-NLS-STOP | Helle Ulrich lab | pNB79; pHU4137 | FRT construct |
| transfected construct (human) | pDEST-BirA*–3FLAG-SMC5 | Helle Ulrich lab | pNB81; pHU4139 | FRT construct |
| transfected construct (human) | pLPC-myc-BirA* | Eros Lazzerini-Denchi lab | pNB176 | Retroviral vector to generate stable cell line |
| transfected construct (human) | pHAGE2-FLAG-SIMC1 | this paper | pNB185 | Lentiviral vector to generate stable cell line |
| transfected construct (human) | pLPC-Myc-BirA*-SIMC1 | this paper | pNB190 | Retroviral vector to generate stable cell line |
| transfected construct (human) | pWZL-FLAG-SLF2 | this paper | pNB195 | Retroviral vector to generate stable cell line |
| transfected construct (human) | pHAGE2 | this paper | pNB248 | Lentiviral vector to generate stable cell line |
| transfected construct (human) | pHAGE2-FLAG-SIMC1 SIM mut | this paper | pNB474 | Lentiviral vector to generate stable cell line |
| transfected construct (human) | pHAGE2-FLAG-SIMC1 combo1 mut | this paper | pNB510 | Lentiviral vector to generate stable cell line; SIMC1 mutations R473D/ N477A/ E480K/ E481K |
| transfected construct (human) | pHAGE2-FLAG-SLF1 | this paper | pNB555 | Lentiviral vector to generate stable cell line |
| antibody | anti-α-tubulin (Mouse monoclonal) | Sigma | T9026 | WB (1:10000) |
| antibody | anti-D tag (Mouse monoclonal) | ABM | G-191 | IF (1:400) |
| antibody | Anti-FLAG (Mouse monoclonal) | Sigma | F3165 | WB (1:5000) |
| antibody | Anti-GFP (Mouse monoclonal) | Santa Cruz | sc-9996 | WB (1:10000) |
| antibody | Anti-phospho-Histone H2A.X (Ser139) (Mouse monoclonal) | Sigma | 05–636 | IF (1:500) |
| antibody | Anti-Myc (Mouse monoclonal) | Scripps Antibody Core Facility | 9E10 | WB (1:3000) |
| antibody | Anti-Myc (Mouse monoclonal) | Invitrogen | MA1-980 | WB (1:2000) |

*Appendix 1 Continued on next page*

*Appendix 1 Continued*

| Reagent type (species) or resource | Designation | Source or reference | Identifiers | Additional information |
|---|---|---|---|---|
| antibody | Anti-PML (Mouse monoclonal) | Santa Cruz | sc-966 | IF (1:200) |
| antibody | Anti-PSTAIR (Mouse monoclonal) | Sigma | P7962 | WB (1:8000) |
| antibody | Anti-PSTAIR (Mouse monoclonal) | Abcam | ab10345 | WB (1:15000) |
| antibody | Anti-SIMC1 (Rabbit Polyclonal) | Abcam | ab241985 | WB (1:1000) |
| antibody | Anti-SMC6 (Rabbit Polyclonal) | Bethyl | A300-237A | IF (1:500) WB (1:1000) |
| antibody | Anti-SUMO2/3 (Mouse monoclonal) | Sigma | MABS2039 | WB (1:1000) |
| antibody | Anti-SV40 LT (Mouse monoclonal) | Abcam | ab16879 | IF (1:400) WB (1:5000) |
| antibody | Goat Anti-Rabbit IgG, HRP (Goat Polyclonal) | Invitrogen | 31460 | WB (1:5000) |
| antibody | Goat Anti-mouse IgG, HRP (Goat Polyclonal) | Invitrogen | 31430 | WB (1:5000) |
| antibody | Goat Anti-mouse IgG2a, Alexa Fluor 488 (Goat Polyclonal) | Life Technologies | A21131 | IF (1:1000) |
| antibody | Goat Anti-rabbit IgG (H+L), Alexa Fluor 555 (Goat Polyclonal) | Life Technologies | A21428 | IF (1:1000) |
| antibody | Goat Anti-mouse IgG2b, Alexa Fluor 555 (Goat Polyclonal) | Life Technologies | A21147 | IF (1:1000) |
| antibody | Goat Anti-mouse IgG, Alexa Flour 647 (Goat Polyclonal) | Jackson Labs | 115-605-006 | IF (1:1000) |
| antibody | Goat Anti-mouse IgG1, Alexa Fluor 647 (Goat Polyclonal) | Life Technologies | A21240 | IF (1:1000) |
| antibody | Goat Anti-mouse IgG2b, Alexa Fluor 647 (Goat Polyclonal) | Life Technologies | A21242 | IF (1:1000) |
| antibody | GFP-Trap magnetic agarose (Alpaca Monoclonal) | ChromoTek | gtma | IP: 25 ul slurry |
| antibody | Myc-Trap magnetic agarose (Alpaca Monoclonal) | ChromoTek | ytma | IP: 25 ul slurry |
| recombinant DNA reagent | pDEST-eGFP-Myc-SLF2 | Helle Ulrich lab | pNB062, pNZ98 | Mammalian Expression vector |
| recombinant DNA reagent | pDEST-eGFP-myc-NLS-STOP | Helle Ulrich lab | pNB68, pNZ110 | Mammalian Expression vector |
| recombinant DNA reagent | pDEST-FLAG-SIMC1 | this paper | pNB133 | Mammalian Expression vector |
| recombinant DNA reagent | pcDNA-H1-sgRNA hSIMC1 exon5 guide 1 | this paper | pNB161 | CRISPR guide |
| recombinant DNA reagent | pcDNA-H1-sgRNA hSIMC1 exon5 guide 2 | this paper | pNB162 | CRISPR guide |

*Appendix 1 Continued on next page*

*Appendix 1 Continued*

| Reagent type (species) or resource | Designation | Source or reference | Identifiers | Additional information |
|---|---|---|---|---|
| recombinant DNA reagent | pLPC-Mre11-GFP | Peiqing Sun lab | pNB168 | Mammalian Expression vector |
| recombinant DNA reagent | pMD2G | Addgene | #12259 | |
| recombinant DNA reagent | psPAX2 | Addgene | #12260 | |
| recombinant DNA reagent | pLPC-FLAG-TIN2 | Eros Lazzerini-Denchi lab | pNB174 | Mammalian Expression vector |
| recombinant DNA reagent | pDEST-FLAG-SIMC1 (284-872) | this paper | pNB175 | Mammalian Expression vector |
| recombinant DNA reagent | pLPC-Myc-SLF2 | this paper | pNB187 | Mammalian Expression vector |
| recombinant DNA reagent | pX330-hSpCas9-puro | Eros Lazzerini-Denchi lab | pNB206 | Mammalian Expression vector, CRISPR |
| recombinant DNA reagent | pcDNA3.1-NACC1-FLAG | GenScript | OHu27779D | |
| recombinant DNA reagent | pEGFP-ZNF451 | **Karvonen et al., 2008** | pNB234 | |
| recombinant DNA reagent | pLPC-Myc-BirA*-SIMC1 (284-872) | this paper | pNB250 | Mammalian Expression vector |
| recombinant DNA reagent | pDEST-mCherry-SIMC1 | this paper | pNB251 | Mammalian Expression vector |
| recombinant DNA reagent | pDEST-mCherry-SLF2 | this paper | pNB256 | Mammalian Expression vector |
| recombinant DNA reagent | pDEST-mCherry-SIMC1-SIM-mut | this paper | pNB261 | SIMC1 SIM mutants FIDL to AADA and VIDL to AADA, corresponding to amino acids 26–29, and 45–48. |
| recombinant DNA reagent | pDEST-eGFP-SIMC1 | this paper | pNB263 | Mammalian Expression vector |
| recombinant DNA reagent | pBabe-SV40LT | Xiaohua Wu lab | pNB312 | Mammalian Expression vector |
| recombinant DNA reagent | pBluescript KS(+)-SV40 | James A. DeCaprio lab | pNB371 | SV40 wild type strain 776 |
| recombinant DNA reagent | pDEST-Flag-NLS-SIMC1 (381-872) | this paper | pNB433 | Mammalian Expression vector |
| recombinant DNA reagent | pDEST-Flag-NLS-SIMC1 (457-872) | this paper | pNB434 | Mammalian Expression vector |
| recombinant DNA reagent | pDEST-Flag-NLS-SIMC1 (652-872) | this paper | pNB435 | Mammalian Expression vector |
| recombinant DNA reagent | pDEST-eGFP-NLS-SLF2 (635–1173) | this paper | pNB437 | Mammalian Expression vector |
| recombinant DNA reagent | pDEST-Flag-NLS-SLF2 (635–1173) | this paper | pNB439 | Mammalian Expression vector |
| recombinant DNA reagent | pACEBac-GST-Tev-SIMC1 (284-872)-PreScission-TwinStrep | this paper | pNB458 | Baculovirus vector |
| recombinant DNA reagent | pACEBac-10xHis-Tev-SLF2 (635–1173) | this paper | pNB468 | Baculovirus vector |
| recombinant DNA reagent | pDEST-eGFP-NLS-SLF1 (410–1058) | this paper | pNB484 | Mammalian Expression vector |

*Appendix 1 Continued on next page*

*Appendix 1 Continued*

| Reagent type (species) or resource | Designation | Source or reference | Identifiers | Additional information |
|---|---|---|---|---|
| recombinant DNA reagent | pDEST-eGFP-NLS-SLF1 (410–725+936-1058) | this paper | pNB485 | Mammalian Expression vector |
| recombinant DNA reagent | pDEST-eGFP-NLS-SLF1 (410–725+GGS + 936-1058) | this paper | pNB486 | Mammalian Expression vector |
| recombinant DNA reagent | pDEST-eGFP-NLS-SLF1 (410-935) | this paper | pNB488 | Mammalian Expression vector |
| recombinant DNA reagent | pDEST-Myc-NLS-SLF2 (635–1173) | this paper | pNB489 | Mammalian Expression vector |
| recombinant DNA reagent | pDEST-GFP-SIMC1 combo1 mut | this paper | pNB497 | Mammalian Expression vector; SIMC1 mutations R473D/ N477A/ E480K/ E481K |
| recombinant DNA reagent | pDEST-Myc-SMC6 | this paper | pNB530 | Mammalian Expression vector |
| recombinant DNA reagent | pDEST-GFP-SIMC1 set A | this paper | pNB578 | Mammalian Expression vector; SIMC1 mutations Y528D/ M529D/ K533D/ L537D |
| recombinant DNA reagent | pDEST-GFP-SIMC1 set B | this paper | pNB579 | Mammalian Expression vector; SIMC1 mutations S778D/ F775E |
| recombinant DNA reagent | pDEST-GFP-SIMC1 set C | this paper | pNB580 | Mammalian Expression vector; SIMC1 mutations L847E/ L850A/ L851E/ Y854D |
| recombinant DNA reagent | pDEST-GFP-NLS-SLF1 (336–1058) | this paper | pNB581 | Mammalian Expression vector |
| recombinant DNA reagent | pDEST-GFP-NLS-SLF1 (336–1058, set A) | this paper | pNB582 | Mammalian Expression vector; SLF1 mutations G464D/ H468E/ A472D/ L476E |
| recombinant DNA reagent | pDEST-GFP-NLS-SLF1 (336–1058, set B) | this paper | pNB583 | Mammalian Expression vector; SLF1 mutations F957E/ L964E |
| recombinant DNA reagent | pDEST-GFP-NLS-SLF1 (336–1058, set C) | this paper | pNB584 | Mammalian Expression vector; SLF1 mutations A1042D/ I1045E/ M1049E |
| recombinant DNA reagent | pDEST-Myc-NLS-SLF2 (635–1173, set A) | this paper | pNB585 | Mammalian Expression vector; SLF2 mutations L775E/ F786E/ F792E |
| recombinant DNA reagent | pDEST-Myc-NLS-SLF2 (635–1173, set B) | this paper | pNB586 | Mammalian Expression vector; SLF2 mutations C820W/ I821W |
| recombinant DNA reagent | pDEST-Myc-NLS-SLF2 (635–1173, set C) | this paper | pNB587 | Mammalian Expression vector; SLF2 mutations M831R/ I835R |
| recombinant DNA reagent | pDEST-Myc-NLS-SLF2 (635–1173, set D) | this paper | pNB588 | Mammalian Expression vector; SLF2 mutations P957D/ V959D/ W1006A |
| sequence-based reagent | sgRNA1_SIMC1_F | this paper | Guide 1 for human SIMC1 CRISPR/Cas9 | AAACgcgttatgtcgttcagaccc |
| sequence-based reagent | sgRNA1_SIMC1_R | this paper | Guide 1 for human SIMC1 CRISPR/Cas9 | GATCCgggtctgaacgacataacgc |
| sequence-based reagent | sgRNA2_SIMC1_F | this paper | Guide 2 for human SIMC1 CRISPR/Cas9 | AAACgggcgagtccttttcctgcg |
| sequence-based reagent | sgRNA2_SIMC1_R | this paper | Guide 2 for human SIMC1 CRISPR/Cas9 | GATCCcgcaggaaaaggactcgccc |

*Appendix 1 Continued*

| Reagent type (species) or resource | Designation | Source or reference | Identifiers | Additional information |
|---|---|---|---|---|
| sequence-based reagent | SIMC1_STOP | this paper | donor template for human SIMC1 CRISPR/Cas9 containing the STOP cassette | TGACTTCTCATTCTTTTTCCCA Cagggacaaactctgcgtgggcga gtcctttc gtgcgGTCGGATCCTTTAAACCTTA ATTAAGCTGTTGTAGttatg tcgttcaga ccctagaagatgactttcagcaga ccctg aggaggcaacggcagca |
| sequence-based reagent | beta actin_F | this paper | qPCR primer | AGGCACCAGGGCGTGAT |
| sequence-based reagent | beta actin_R | this paper | qPCR primer | GCCCACATAGGAATCCTTCTGAC |
| sequence-based reagent | SIMC1 3′UTR_F | this paper | qPCR primer | cctgccaagcactgaatgcc |
| sequence-based reagent | SIMC1 3′UTR_R | this paper | qPCR primer | ccatatttgagaacaggctaggatagg |
| peptide, recombinant protein | Benzonase | EMD Millipore | 70746 | |
| peptide, recombinant protein | cOmplete, Mini, EDTA-free Protease Inhibitor Cocktail Tablets | Sigma | 4693159001 | |
| peptide, recombinant protein | Dynabeads MyOne Streptavidin C1 | ThermoFisher | 65001 | |
| peptide, recombinant protein | Dynabeads MyOne Streptavidin T1 | ThermoFisher | 65601 | |
| peptide, recombinant protein | Endoproteinase LysC | New England BioLabs | P8109S | |
| peptide, recombinant protein | Glutathione Agarose Resin | GoldBio | G-250–100 | |
| peptide, recombinant protein | HaltTM Protease Inhibitor Cocktail, EDTA-Free (100 X) | ThermoFisher | 78437 | |
| peptide, recombinant protein | protease inhibitor cocktail | Pierce | A32963 | |
| peptide, recombinant protein | Tev protease | this paper | | prepared in the lab |
| peptide, recombinant protein | Trypsin Protease | Pierce | 90057 | |
| chemical compound, drug | arginine-0/lysine-0 (light) | Sigma | A6969 and L8662 | |
| chemical compound, drug | arginine-6/lysine-8 (heavy) | Cambridge Isotope Laboratories | CNLM-539-H-1 and CNLM-291-H-1 | |
| chemical compound, drug | Biotin | Sigma | B4501 | |

*Appendix 1 Continued on next page*

*Appendix 1 Continued*

| Reagent type (species) or resource | Designation | Source or reference | Identifiers | Additional information |
|---|---|---|---|---|
| chemical compound, drug | Cold water fish gelatin | Sigma | G7765 | |
| chemical compound, drug | 4′,6-diamidino-2-phenylindole dihydrochloride (DAPI) | Sigma | D-9542 | |
| chemical compound, drug | DMEM for SILAC | ThermoFisher | 88364 | |
| chemical compound, drug | Dimethyl sulfoxide (DMSO) | Sigma | D8418 | |
| chemical compound, drug | Gelatin (type A) | MP bio | 901771 | |
| chemical compound, drug | Hoechst 33342 | Invitrogen | H3570 | |
| chemical compound, drug | Hygromycin B | Invivogen | ant-hg | |
| chemical compound, drug | Iodoacetamide | Sigma | | |
| chemical compound, drug | Lipofectamine 2000 | ThermoFisher | 11668019 | |
| chemical compound, drug | MMS | Sigma | 129925 | |
| chemical compound, drug | Nocodazole | Fisher Scientific | NC1084348 | |
| chemical compound, drug | Normal goat serum | BioLegend | 927502 | |
| chemical compound, drug | Polyethylenimine (PEI) | Polysciences | 02371–500 | |
| chemical compound, drug | ProBond Ni affnity resin | Invitrogen | 46–0019 | |
| chemical compound, drug | Superdex 200 10/300 | GE Healthcare | 17517501 | |
| chemical compound, drug | TCEP | Pierce | 20490 | |
| chemical compound, drug | TransIT-LT1 | Mirus | MIR2300 | |
| chemical compound, drug | Triton X-100 | Sigma | T9284 | |
| chemical compound, drug | Trypsin EDTA 0.25% | ThermoFisher | 25300054 | |
| chemical compound, drug | UltraAuFoil R1.2/1.3 300-mesh grids | Quantifoil | | |
| commercial assay or kit | CometAssay | Trevigen | 4250–050 K | |
| commercial assay or kit | In-Fusion HD Cloning Plus Kits | Takara | 638910 | |
| commercial assay or kit | NuPAGE LDS Sample Loading Buffer | ThermoFisher | NP0007 | |
| commercial assay or kit | ProLong Gold Antifade | Invitrogen | P36934 | |
| commercial assay or kit | Rneasy Plus Mini Kit | QIAGEN | 74104 | |

*Appendix 1 Continued on next page*

*Appendix 1 Continued*

| Reagent type (species) or resource | Designation | Source or reference | Identifiers | Additional information |
|---|---|---|---|---|
| commercial assay or kit | Sensifast SYBR No-rox | Meridian Bioscience | BIO-98020 | |
| commercial assay or kit | Superscript III First-Strand Synthesis System for RT-PCR | Invitrogen | 18080–051 | |
| commercial assay or kit | SuperSignal West Dura Extended Duration Substrate | ThermoFisher | 34076 | |
| commercial assay or kit | West-Q Pico ECL solution | genDEPOT | W3652-020 | |
| software, algorithm | AlphaFold 2 | https://doi.org/10.1038/s41586-021-03819-2 | | |
| software, algorithm | AlphaFold-Multimer | https://doi.org/10.1101/2021.10.04.463034 | | |
| software, algorithm | ColabFold | https://doi.org/10.1038/s41592-022-01488-1 | | |
| software, algorithm | HHpred | https://doi.org/10.1016/j.jmb.2017.12.007 | | |
| software, algorithm | IP2 - Integrated Proteomics Pipeline | https://doi.org/10.1016/j.jsb.2009.01.002 | | |

