## [Editor Report]

This paper will be of interest to the chromosome biology field and SMC researchers in particular. The study provides cell biological, biochemical, and structural modeling evidence that a new Nse5-like protein named SIMC1 is a paralog of SLF1, and that the two compete for SLF2-Smc5/6 binding. The authors also show that SIMC1 targets SMC5/6 to polyomavirus replication centers through its SUMO binding motifs (SIMs), while SIMC1 is not recruited to DNA damage sites, supporting a specific role for SIMC1 in Smc5/6 recruitment for viral restriction.

---

## [Decision Letter]

**Decision letter after peer review:**

Thank you for submitting your article "The Nse5/6-like SIMC1-SLF2 Complex Localizes SMC5/6 to Viral Replication Centers" for consideration by *eLife*. Your article has been reviewed by 3 peer reviewers, and the evaluation has been overseen by a Reviewing Editor and Jessica Tyler as the Senior Editor. The reviewers have opted to remain anonymous.

Essential revisions:

1) The cryoEM structure is at relatively low resolution, and should therefore be verified by mutagenesis. Specifically, to support the structural data and the conclusion that similar SIMC1 and SLF1 surfaces bind SLF2, the authors should mutate surface residues between SIMC1 and SLF2, and confirm that these mutations interfere with SIMC1-SLF2 interaction.

2) The claims about the exclusive roles of SIMC1-SLF2 and SLF1-SLF2 need to be better substantiated. Specifically, experiments directly showing that SLF1 is not recruited to LT-induced foci and PML bodies, and that SIMC1 is not recruited to damage foci, should be provided. Similarly, is SLF1:2 the predominant form of the SLF2-containing complex, while SIMC1:SLF2 is specifically induced by LT transformation?

3) The SIMC1 combo mutant might affects the interaction between SIMC1-SLF2 and Smc5/6. A CoIP should be conducted to test this, providing a better understanding of the shown defect in Smc5/6 recruitment.

4) More information on the structural similarities/differences between SIMC1 and SLF1 should be provided with text revisions. For example, are the conserved surface regions described for SIMC1 (Figure 6A) also seen in the SLF1/2 structure model? There appears to be a loop in SIMC1 involved in dimerization with SLF2, is this a conserved feature in SLF1? Is the SLF1:2 interface also formed by positively charged patches as seen in the SIMC1:SLF2 model? Are SIMs unique for SIMC1 or are they also found in SLF1? In addition, to support the claim that the "structure of the SIMC1-SLF2 dimer resembles the yeast NSE5-NSE6 dimer", the overlap between the structures of the dimers should be shown.

Other concerns can be addressed by toning down the conclusions. For example, the current experiments indicate proximity (not interaction) between SIMC1 and ZNF451 (Reviewer 1, point 4).

*Reviewer #1 (Public Review):*

The authors used BioID to identify SMC5 proximal proteins in HEK293T cells. In one of the top hits, they recognized an NSE5-like domain and proposed it (SIMC1) to be the new NSE5-like subunit of the human SMC5/6 complex. Accordingly, they showed its colocalization and co-immunoprecipitation with the NSE6/SLF2 subunit. They confirmed SIMC1-SLF2 direct physical interaction by their copurification. Furthermore, using cryoEM and AlphaFold modelling, they were able to determine the structure of the SIMC1-SLF2 dimer at 3.9A resolution. They also showed that the SIMC1-SLF2 dimer is mutually exclusive to the previously described SLF1-SLF2 dimer. To make their structural data more solid, the authors should have mutated contact residues to support their structural data and their conclusion that the similar SIMC1 and SLF1 surfaces bind SLF2. Otherwise, the above data are very strong and support the author's conclusions about the new NSE5-like subunit of the human SMC5/6 complex.

Further, the authors showed that the SMC5/6 complex is localized to SV40 T-large (LT) antigen-induced foci and PML bodies. This localization is dependent on the SIMC1 and its SUMO-interacting motifs. In addition, several SUMO pathway factors (known components of PML bodies) were identified in the SIMC1 BioID search. These data suggest a strong connection of SMC5/6 to SUMO-rich sites. Based on this connection, the authors speculate that SIMC1-SLF2 specifically regulates responses to viral challenges while SLF1-SLF2 is specifically involved in DNA damage responses. Given the roles of PML bodies in homologous recombination (DNA damage repair, alternative telomere lengthening), their experiments do not directly prove this assumption. Actually, SMC5/6 was already shown to be localized to PML bodies and involved in alternative telomere lengthening (https://pubmed.ncbi.nlm.nih.gov/17589526/). In conclusion, further experiments are required to show the exclusive roles of distinct SLF2 complexes and the specific involvement of SIMC1-SLF2 in antiviral responses.

Major requests:

1. The biochemical data on the SIMC1-SLF2 dimer are solid. However, the authors did not verify the structural data by mutating surface residues mediating SIMC1-SLF2 interaction (although they prepared various constructs and generated "combo" mutations at the different surfaces – Figure 7). They should mutate contact residues to support their structural data and their conclusion that the same SIMC1/SLF1 surfaces bind SLF2.

2. The claims about the exclusive roles of SIMC1-SLF2 and SLF1-SLF2 should be better substantiated. The control experiment showed (no)localization of SLF1 to LT-induced foci and PML bodies (and vice versa SIMC1 (no)localization in DNA-damaged cells like in Raschle, 2015 – https://pubmed.ncbi.nlm.nih.gov/25931565/) should be performed.

3. In addition, the sensitivity of the SIMC1-/- cells to replication stress and DNA damage agents should be compared to the other SMC5/6 mutants.

4. SIMC1 BioID (under native/denaturing conditions) experiment suggests that multiple SUMO pathway factors are proximal to SIMC1. However, there is no evidence for the interaction between SIMC1 and ZNF451 (p. 12/line 228 + p.28/line 508). The coIP experiment (Figure 3D) was performed in a biotin-streptavidin manner suggesting the proximity of these proteins but not the interaction. The coIP experiment should be performed with anti-myc or anti-GFP antibodies.

*Reviewer #1 (Recommendations for the authors):*

1. In the BioID search for SMC5 proximal proteins, all but NSE5/SLF1 and NSE2 subunits of the SMC5/6 complex were detected. NSE5/SLF1 and NSE2 subunits were missing either for a technical reason or because SIMC1 replaced NSE5/SLF1 in PML/LT foci. Does this data mean that this PML-proximal SMC5/6 complex is also missing the NSE2 SUMO-ligase subunit, then?

2. Results from Potts et al. (https://pubmed.ncbi.nlm.nih.gov/17589526/, suggesting NSE2 localization to PML bodies should be cited.

3. Under native BioID conditions, both biotinylated proteins and proteins associated with the biotinylated proteins are precipitated, while only biotinylated proteins are precipitated under denaturing BioID conditions (Figure 3). It suggests that SIMC1 is not in close proximity with SMC5/6 core subunits (It is a bit misleading to conclude the paragraph with "SIMC1 is an SMC5/6 subunit"?; p. 12/line 213).

4. The "combo" mutant disturbed the localization of SMC6 to LT-containing foci while SIMC1 localization remains intact. This suggests that SIMC1 interaction with a core SMC5/6 subunit is disturbed (which contradicts the conclusion from BioID experiment that SIMC1 is not binding to the SMC5/6 core). How do the authors interpret this result?

5. It is not clear how does "The ellipsoid-like overall shape of the SIMC1-SLF2 complex suggests that the complex likely uses its surface to regulate SMC5/6 function"? (p. 22/line 385).

*Reviewer #2 (Public Review):*

In this manuscript, the authors provide cell biological, biochemical, and structural modeling data to suggest that SIMC1 is a paralog of SLF1 and that the two compete for SLF2 binding likely because their C-terminal regions form similar helix-enriched structures to bind at the same surface of SLF2. As previously shown for SLF1, the authors found that SIMC1 bound to other subunits of Smc5/6. Distinct from SLF1, SIMC1 contains N-terminal SIMs and binds to SUMO pathway proteins including PML and SUMO. They found that SIMs and the SLF2-binding regions of SIMC1 are both required for targeting SIMC1 and SMC5/6 to the PML body upon LT expression. The authors showed that structural models of the dimerized portions of SIMC1:SLF2 and SLF1:SLF2 are similar to that of the yeast Nse5-6 structures, suggesting that they are Nse5-6 homologs but with different roles in targeting the SMC5/6 complex.

Overall, this is a very interesting study. The experiments were well done and data are presented clearly. The report will be highly interesting to the SMC field.

*Reviewer #2 (Recommendations for the authors):*

Overall, this is a very interesting study. The experiments were well done and data are presented clearly. The report will be highly interesting to the SMC field and is well suited to be published in *eLife*. I would like to congratulate the authors on the advances reported in this work. I only have a couple of questions/suggestions for the authors to consider.

In figure 6, the authors examined a surface area of SIMC1 that has a high conservation score. A combo mutant affecting four residues in this area reduced SMC6 foci levels within the PyVRCs. The reason for this effect is unclear but could be due to impairment of association with Smc6. This straightforward test should be conducted to provide an understanding of the shown defect.

The similarities and differences between the two Nse5 paralogs need more clarification. Given the structural models have already been established, a few specifications should be quite feasible to add. For example, are the conserved surface regions described for SIMC1 (Figure 6A) seen in the SLF1/2 structure model or conserved surfaces are distinct between the two complexes? There appeared to be a loop in SIMC1 involved in dimerization with SLF2, is this a conserved feature in SLF1? Is the SLF1:2 interface also formed by positively charged patches as seen in the SIMC1:SLF2 model? Whether SIMs are unique for SIMC1 or are also found in SLF1.

Given the model proposed in this work, one would predict that SLF1:2 is the predominant form of the SLF2-containing complex, while SIMC1:SLF2 is induced by LT transformation. Do the authors have data to support this idea?

*Reviewer #3 (Public Review):*

In yeasts *Saccharomyces cerevisiae* and *Schizosaccharomyces pombe*, the Smc5/6 complex is associated with regulatory subunits Nse5 and Nse6, which have been shown to be required for chromatin association and also inhibit ATP turnover. In addition, the recruitment to sites of DNA damage requires a multi-BRCT domain protein, Brc1/Rtt107. The identity of the Nse5 and Nse6 regulatory subunits of the human SMC5/6 complex has long been a mystery. The Nse6 homologue, SLF2, was identified in a proteomics screen for factors required for bypass of DNA crosslinks. SLF1, the other recruiter identified in the same screen, showed very limited homology to Nse6 and also contained BRCT domains and ANKRIN repeats suggesting that it provided an analogous mechanism of recruitment of SMC5/6 to sites of DNA damage to that of Brc1/Rtt107. The questions of whether SLF1 was the functional homologue of Nse5 in humans and whether it was required for all chromatin association remained.

The authors address this in their analysis. They used proteomics to identify SMC5/6 interactors in PML nuclear bodies. They identified a novel regulator, SIMC1, that contains SUMO interacting motifs (SIMs) and an Nse5-like domain. They showed that SIMC1 and SLF2 interact through the Nse5 domain and form an alternative complex to SLF1/SLF2. They then suggested that the SIMC1/SLF2 complex is specific for recruitment of SMC5/6 for restriction of viral replication/transcription as the SIM and Nse5 domains were found to be important to localise SMC5/6 to polyomavirus replication centres at PML bodies, which are enriched for SUMO, and for recruitment of SMC5/6 to SV40 replication centres. Since PML bodies are associated with alternative lengthening of telomeres (ALT) and SMC5/6 has roles in telomere maintenance it would be interesting to know whether the SIMC1/SLF2 complex was also required in these circumstances but this is a question for a future study.

Overall, the manuscript is well written and the data is of high quality. It makes a step advance in terms of our understanding of how the Smc5/6 complex functions.

---

## [Author Response]

Essential revisions:1) The cryoEM structure is at relatively low resolution, and should therefore be verified by mutagenesis. Specifically, to support the structural data and the conclusion that similar SIMC1 and SLF1 surfaces bind SLF2, the authors should mutate surface residues between SIMC1 and SLF2, and confirm that these mutations interfere with SIMC1-SLF2 interaction.

We have performed mutational analyses on the interface residues of SIMC1 and SLF2. The results presented in Figure 5 —figure supplement 14 show that the mutations indeed disrupt the SIMC1-SLF2 binding, thereby supporting the structure. We also performed similar analyses on the predicted SLF1/2 structure and obtained results supporting the prediction (Figure 7 —figure supplement 8).

2) The claims about the exclusive roles of SIMC1-SLF2 and SLF1-SLF2 need to be better substantiated. Specifically, experiments directly showing that SLF1 is not recruited to LT-induced foci and PML bodies, and that SIMC1 is not recruited to damage foci, should be provided. Similarly, is SLF1:2 the predominant form of the SLF2-containing complex, while SIMC1:SLF2 is specifically induced by LT transformation?

We have included new data showing that as previously observed (Raschle et al., Science 2015) SLF1 is recruited to laser damage stripes that overlap with γ-H2A. However, unlike SLF1, SIMC1 shows no significant localization to laser induced DNA damage stripes (New Figure 8).

Interestingly, overexpressed SLF1 localizes proximal to LT sites in a manner dependent on the presence of SIMC1 (New Figure 8). This may be due to the presence of SLF2 at LT sites in the presence of SIMC1, which could bind overexpressed SLF1. In any event, SMC5/6 does not significantly localize to LT foci in the absence of SIMC1, demonstrating the dominant role of SIMC1 in targeting PML bodies/viral replication sites. This is also consistent with the presence of SIMs in SIMC1 but not SLF1. Moreover, SLF1 was not isolated from LT foci in our SMC5 BioID, suggesting that overexpression may be driving its unusual localization.

In addition, two recent studies we include citations and discussion of in the revised manuscript show that viral restriction by SMC5/6 is SLF1-independent. Finally, another recent genetic study found that SIMC1 is required for rAAV restriction, although no further functional analyses were performed. These findings, together with ours, make it abundantly clear that SIMC1 is a viral specific Nse5-like protein in human cells.

The test of which complex is predominant would require antibodies to both SLF1/SIMC1, which are not currently available. However, our qPCR analysis in supplementary data shows that the levels of SIMC1 and SLF1 transcripts are not affected differently by LT expression (New Figure 8 —figure supplement 1).

3) The SIMC1 combo mutant might affects the interaction between SIMC1-SLF2 and Smc5/6. A CoIP should be conducted to test this, providing a better understanding of the shown defect in Smc5/6 recruitment.

We have added new data showing that the combo mutant of SIMC1 is indeed deficient in SMC6 binding (New Figure 6b). This likely explains the defect in SMC5/6 recruitment by the SIMC1 combo mutant, despite the mutant’s localization at LT foci.

4) More information on the structural similarities/differences between SIMC1 and SLF1 should be provided with text revisions. For example, are the conserved surface regions described for SIMC1 (Figure 6A) also seen in the SLF1/2 structure model? There appears to be a loop in SIMC1 involved in dimerization with SLF2, is this a conserved feature in SLF1? Is the SLF1:2 interface also formed by positively charged patches as seen in the SIMC1:SLF2 model? Are SIMs unique for SIMC1 or are they also found in SLF1? In addition, to support the claim that the "structure of the SIMC1-SLF2 dimer resembles the yeast NSE5-NSE6 dimer", the overlap between the structures of the dimers should be shown.

We have performed more analyses on the SLF1/2 model and compared SIMC1 and SLF1. Conservation analysis of SLF1 surface shows that the SLF2 interacting interface contains conserved residues while the conservation scores appear to be less than those of SIMC1 (Figure 7 —figure supplement 7). This may suggest SLF1 sequences are more diverged across various species compared to SIMC1. The SLF1 residues that correspond to the SIMC1 combo residues are not identical but similar to SIMC1’s residues. There is one identical residue (Figure 7 —figure supplement 9). The N-terminal region to the α-solenoid of SLF1 forms helical conformation, and like SIMC1, it is inserted into the space between SLF1 and SLF2 (Figure 7 —figure supplement 6). The SLF1 interface is also charged, and the charges appear to be involved in the dimerization (Figure 7 —figure supplement 7). We have also superimposed the yeast Nse5/6 complex onto the SIMC1-SLF2 complex (Figure 5f), which supports the similarity between them and reveals the differences.

Other concerns can be addressed by toning down the conclusions. For example, the current experiments indicate proximity (not interaction) between SIMC1 and ZNF451 (Reviewer 1, point 4).

We have updated the text describing the proximity versus interaction of SIMC1 and ZNF451.

Reviewer #1 (Public Review):The authors used BioID to identify SMC5 proximal proteins in HEK293T cells. In one of the top hits, they recognized an NSE5-like domain and proposed it (SIMC1) to be the new NSE5-like subunit of the human SMC5/6 complex. Accordingly, they showed its colocalization and co-immunoprecipitation with the NSE6/SLF2 subunit. They confirmed SIMC1-SLF2 direct physical interaction by their copurification. Furthermore, using cryoEM and AlphaFold modelling, they were able to determine the structure of the SIMC1-SLF2 dimer at 3.9A resolution. They also showed that the SIMC1-SLF2 dimer is mutually exclusive to the previously described SLF1-SLF2 dimer. To make their structural data more solid, the authors should have mutated contact residues to support their structural data and their conclusion that the similar SIMC1 and SLF1 surfaces bind SLF2. Otherwise, the above data are very strong and support the author's conclusions about the new NSE5-like subunit of the human SMC5/6 complex.Further, the authors showed that the SMC5/6 complex is localized to SV40 T-large (LT) antigen-induced foci and PML bodies. This localization is dependent on the SIMC1 and its SUMO-interacting motifs. In addition, several SUMO pathway factors (known components of PML bodies) were identified in the SIMC1 BioID search. These data suggest a strong connection of SMC5/6 to SUMO-rich sites. Based on this connection, the authors speculate that SIMC1-SLF2 specifically regulates responses to viral challenges while SLF1-SLF2 is specifically involved in DNA damage responses. Given the roles of PML bodies in homologous recombination (DNA damage repair, alternative telomere lengthening), their experiments do not directly prove this assumption. Actually, SMC5/6 was already shown to be localized to PML bodies and involved in alternative telomere lengthening (https://pubmed.ncbi.nlm.nih.gov/17589526/). In conclusion, further experiments are required to show the exclusive roles of distinct SLF2 complexes and the specific involvement of SIMC1-SLF2 in antiviral responses.Major requests:1. The biochemical data on the SIMC1-SLF2 dimer are solid. However, the authors did not verify the structural data by mutating surface residues mediating SIMC1-SLF2 interaction (although they prepared various constructs and generated "combo" mutations at the different surfaces – Figure 7). They should mutate contact residues to support their structural data and their conclusion that the same SIMC1/SLF1 surfaces bind SLF2.

As mentioned above, our new mutagenesis data support our structure of SIMC1-SLF2 complex and the AlphaFold-predicted model of the SLF1/2 complex, thereby solidifying our conclusion that SIMC1 and SLF1 compete on the same surface of SLF2. The new results are shown in Figure 5 —figure supplement 14, and Figure 7 —figure supplement 8.

2. The claims about the exclusive roles of SIMC1-SLF2 and SLF1-SLF2 should be better substantiated. The control experiment showed (no)localization of SLF1 to LT-induced foci and PML bodies (and vice versa SIMC1 (no)localization in DNA-damaged cells like in Raschle, 2015 – https://pubmed.ncbi.nlm.nih.gov/25931565/) should be performed.

This has been addressed and the results described above in the “essential revisions” section.

3. In addition, the sensitivity of the SIMC1-/- cells to replication stress and DNA damage agents should be compared to the other SMC5/6 mutants.

We only have SIMC1 null HEK293 cells, which are not ideal for DNA damage studies. SIMC1 shRNA is unfortunately not effective enough to assay other cell lines and CRISPR editing has not been successful in other cell lines. Importantly, we have shown that SIMC1 does not localize to laser-induced DNA damage stripes (New Figure 8), strongly suggesting it does not regulate the DNA repair functions of SMC5/6.

4. SIMC1 BioID (under native/denaturing conditions) experiment suggests that multiple SUMO pathway factors are proximal to SIMC1. However, there is no evidence for the interaction between SIMC1 and ZNF451 (p. 12/line 228 + p.28/line 508). The coIP experiment (Figure 3D) was performed in a biotin-streptavidin manner suggesting the proximity of these proteins but not the interaction. The coIP experiment should be performed with anti-myc or anti-GFP antibodies.

We did not intend to indicate that this was a direct interaction. We have changed the text to make it clear that this is a proximal interaction, as for other hits in our BioID screen except SLF2.

Reviewer #1 (Recommendations for the authors):1. In the BioID search for SMC5 proximal proteins, all but NSE5/SLF1 and NSE2 subunits of the SMC5/6 complex were detected. NSE5/SLF1 and NSE2 subunits were missing either for a technical reason or because SIMC1 replaced NSE5/SLF1 in PML/LT foci. Does this data mean that this PML-proximal SMC5/6 complex is also missing the NSE2 SUMO-ligase subunit, then?

NSE5/SLF1 was missing from the purification likely due to the presence of LT in HEK293T cells, which would have enriched for the SMC5/6 SIMC1/SLF2 complex. The absence of NSE2 may be due to the limited biotinylation radius of BirA* used in our purifications (~10 nm). Given that SMC5 is a “rod-like” structure of ~50 nm it is possible that the SMC5-BirA* biotinylation did not reach NSE2 that binds ~ 20 nm away from SMC5-BirA*.

2. Results from Potts et al. (https://pubmed.ncbi.nlm.nih.gov/17589526/, suggesting NSE2 localization to PML bodies should be cited.

The reference has been added.

3. Under native BioID conditions, both biotinylated proteins and proteins associated with the biotinylated proteins are precipitated, while only biotinylated proteins are precipitated under denaturing BioID conditions (Figure 3). It suggests that SIMC1 is not in close proximity with SMC5/6 core subunits (It is a bit misleading to conclude the paragraph with "SIMC1 is an SMC5/6 subunit"?; p. 12/line 213).

This is likely for similar reasons as the absence of NSE2 in the SMC5 purification, that is, a narrow radius of biotinylation. NSE5/6 are considered regulatory SMC5/6 subunits, so then should SIMC1/SLF2. Nevertheless, we have adjusted the text to describe SIMC1 as an NSE5-like cofactor of SMC5/6.

4. The "combo" mutant disturbed the localization of SMC6 to LT-containing foci while SIMC1 localization remains intact. This suggests that SIMC1 interaction with a core SMC5/6 subunit is disturbed (which contradicts the conclusion from BioID experiment that SIMC1 is not binding to the SMC5/6 core). How do the authors interpret this result?

Again, the BioID data reflect a very narrow biotinylation radius ~10 nm. The architecture of the complex and placement of the BirA* tag may affect which proteins are identified. We now show that the SIMC1 combo mutant fails to interact with SMC6 using co-IP, which neatly explains the defect in SMC5/6 recruitment despite its localization at LT sites (New Figure 6b).

5. It is not clear how does "The ellipsoid-like overall shape of the SIMC1-SLF2 complex suggests that the complex likely uses its surface to regulate SMC5/6 function"? (p. 22/line 385).

As it was not a polished sentence and not crucial for the narrative, we omitted this sentence in the revision.

Reviewer #2 (Public Review):In this manuscript, the authors provide cell biological, biochemical, and structural modeling data to suggest that SIMC1 is a paralog of SLF1 and that the two compete for SLF2 binding likely because their C-terminal regions form similar helix-enriched structures to bind at the same surface of SLF2. As previously shown for SLF1, the authors found that SIMC1 bound to other subunits of Smc5/6. Distinct from SLF1, SIMC1 contains N-terminal SIMs and binds to SUMO pathway proteins including PML and SUMO. They found that SIMs and the SLF2-binding regions of SIMC1 are both required for targeting SIMC1 and SMC5/6 to the PML body upon LT expression. The authors showed that structural models of the dimerized portions of SIMC1:SLF2 and SLF1:SLF2 are similar to that of the yeast Nse5-6 structures, suggesting that they are Nse5-6 homologs but with different roles in targeting the SMC5/6 complex.Overall, this is a very interesting study. The experiments were well done and data are presented clearly. The report will be highly interesting to the SMC field.Reviewer #2 (Recommendations for the authors):Overall, this is a very interesting study. The experiments were well done and data are presented clearly. The report will be highly interesting to the SMC field and is well suited to be published in eLife. I would like to congratulate the authors on the advances reported in this work. I only have a couple of questions/suggestions for the authors to consider.In figure 6, the authors examined a surface area of SIMC1 that has a high conservation score. A combo mutant affecting four residues in this area reduced SMC6 foci levels within the PyVRCs. The reason for this effect is unclear but could be due to impairment of association with Smc6. This straightforward test should be conducted to provide an understanding of the shown defect.

This data has been included (Figure 6b) and supports your prediction that this SIMC1 surface binds SMC6. The combo mutant still interacts with SLF2 and localizes at PyVRCs. It is therefore likely that the defect in SMC6 contact by the SIMC1 combo mutant leads to the loss of SMC6 recruitment to PyVRCs.